# Patient perceived weight stigma and patient-centered language use preferences: A cross-sectional mixed methods analysis conducted in a large academic medical center

**Ryan M. Kane**[1,2,3]*, **Selvi B. Williams**[1], **Kimberly Reynolds**[4], **Abby Kincanon**[5], **Marcy R. Hager**[6], **Craig McDougall**[1], **Jonathan Q. Purnell**[7], **Patricia A. Carney**[8]

1 Department of Medicine, Oregon Health & Science University School of Medicine, Portland, OR, United States of America, 2 Division of General Internal Medicine, Department of Medicine, Duke University School of Medicine, Durham, NC, United States of America, 3 Clinical and Translational Science Institute, Duke University, Durham, NC, United States of America, 4 Department of Pediatrics, Oregon Health & Science University School of Medicine, Portland, OR, United States of America, 5 NRC Health, Lincoln, NE, United States of America, 6 Office of Clinical Integration and Evidence-based Practice, Oregon Health & Sciences University, Portland, OR, United States of America, 7 Division of Cardiovascular Medicine, Knight Cardiovascular Institute, Oregon Health & Science University School of Medicine, Portland, OR, United States of America, 8 Department of Family Medicine, Oregon Health & Science University School of Medicine, Portland, OR, United States of America

* ryan.kane@duke.edu

## Abstract

### Background

Due to the rising prevalence of obesity and its impact on healthcare, patient-specific context is needed to optimize weight management with an emphasis on reducing health care-associated weight stigma. Our survey aimed to explore institution-specific patient experiences of weight stigma and preferences for patient-centered language use regarding weight management care.

### Methods

This cross-sectional analysis adopted a concurrent mixed methods design with a sample of individuals who opted in to complete patient experience surveys after receiving care at a large academic medical center in the United States (U.S.). Categorical and continuous variables were assessed using Chi-squared and analysis of variance. We used classical content analysis to qualitatively analyze free-text data for thematic coding.

### Results

After a 1-week survey fielding period, 3,219 of 16,758 patients completed the survey, yielding a response rate of (19.2%) with 2,816 having available electronic health record body mass index (BMI) data. Patients were comfortable discussing weight with their primary care providers but showed variation in the preferred approach and terms. Female patients with higher BMIs reported higher rates of delayed and canceled care due to prior weight stigma

**Data Availability Statement:** Data cannot be publicly shared due to OHSU's IRB approval process restrictions. De-identified data can be made available for validation purposes upon request for researchers who meet criteria for access to confidential data, according to OHSU's IRB policies. Please contact the OHSU IRB at IRB@ohsu.edu to request additional information for data validation purposes.

**Funding:** The author(s) received no specific funding for this work.

**Competing interests:** The authors have declared that no competing interests exist.

(25.6% and 12.2% for patients with class 3 obesity), and preferred a slower, gentler, and less direct approach with term preferences for "healthy eating plan" and against "obesity." Qualitative analysis yielded 27 themes grouped into three domains: Emotional Hinderances, Perceptual Hinderances, and Perceived Helpfulness.

## Conclusions

Findings from our large single institution cohort expand on the existing weight stigma literature by identifying patient language preferences and healthcare experiences according to patient weight class and sex. Given the potential impact of understanding context-specific patient language use preferences to reduce weight stigma, we recommend other healthcare systems use a similar process to address weight stigma as part of a coordinated health system improvement initiative to enhance patient-centered weight management care.

## Introduction

Globally, the prevalence of obesity has nearly tripled since 1975 [1]. A recent U.S. survey of adults over age 20 years old (2017–2018) found the prevalence of obesity has reached 42.4% and, when combined with overweight, was 73.6% of the adult population [2]. Managing obesity as chronic disease is challenging due to its multifactorial nature—influenced by individual factors (e.g., genetics and co-occurring medical conditions), psychological factors (e.g., weight stigma and behaviors), and societal factors (e.g., social drivers of health) [2–5]. Given this concerning trend and treatment complexity, the National Academies of Sciences, Engineering, and Medicine and World Health Organization have recommended multi-level, multi-modal approaches for healthcare institutions to work within their individual clinical and community environments to enhance the prevention of and treatment for obesity, using a chronic care model [3, 6, 7]. The chronic care model highlights critical elements for health care systems to enhance the provision of high-quality chronic disease care across patient, health system, and community spheres of influence [8].

These interventions often begin with patient-clinician or patient-health system interactions. As compellingly summarized by Albury, et al.'s 2020 joint consensus statement, studies assessing patient and provider communication when discussing weight management have revealed that many patients with obesity experience explicit and implicit stigma when interacting with healthcare providers and systems [9]. Moreover, a recent review modeled how negative provider attitudes and behaviors can strain the patient-clinician relationship, resulting in adverse patient outcomes (e.g. delayed/avoidant care seeking behaviors, decreased adherence to physician recommendations, and a lower likelihood of weight loss) [10, 11]. These negative provider attitudes and behaviors include (1) enacted stigma (behaviors resulting from negative attitudes like decreased patient-centered communication due to providers' belief that patients with obesity are less likely to adhere to treatments), (2) threatening environmental cues (clinical spaces and equipment that are not accommodating to patients of all sizes), and stereotypical behaviors (providers demonstrating less respect towards patients with obesity due to perceptions of laziness) [10]. Review of qualitative analyses identified related emergent themes surrounding (1) verbal/non-verbal communication of stigma, (2) the impact of weight stigma upon care provision, and (3) weight stigma's impact upon systemic barriers to care [12]. Clinical encounters are particularly important because they can alleviate or aggravate already internalized

weight stigma and promote or antagonize a patient's efforts toward making positive lifestyle changes or complying with anti-obesity medications when indicated as part of a comprehensive weight-management program [9]. Moreover, weight stigma has deleterious effects for patient engagement in preventive care, healthcare avoidance, biased physician decision making, poor patient-provider communication, and increased primary care switching [10, 13–16]. Given the variability in contextual factors, healthcare systems should conduct their own internal assessments to identify attitudes, behaviors, and institutional practices that both hinder and facilitate patients' weight management. This information can then inform future health system improvements in the provision of compassionate, equitable, and effective weight management care.

At our large academic medical center located in the Pacific Northwest of the U.S., we convened a diverse group of partners to engage in a system-wide institutional review to develop and implement an evidence-based clinical guideline to address weight management for pediatric and adult patients [17, 18]. Given the adverse effects of health care-associated weight stigma, we first sought to understand how patients with overweight or obesity in our health system perceived their previous care experiences. Though prior literature demonstrated weight stigma-focused analyses of specific cohorts, we were unaware of prior health systems who had published a similar institutional approach to better understand the direct language use preferences and care experiences of their own patient population. To assess this in our health system, we conducted a cross-sectional survey with an existing institutional group of patients through our organization's contracted survey vendor (NRC Health). The primary aim of this project was to understand patient experiences at our institution, with a specific focus on patient weight stigma and their preferences for patient-centered language use in communication about obesity care, to inform future health system improvement.

## Methods

### Study setting and survey development

This survey was conducted at Oregon Health & Science University (OHSU), which has schools of dentistry, medicine, nursing, pharmacy, and public health. It has a 576-bed teaching hospital with biomedical research facilities, a Level I trauma center, and a primary care population of over 100,000 patients.

The survey itself developed from our initial institutional weight management guideline development process, which sought to improve the care for people living with overweight and obesity through developing guidelines, standards, and policies that promote optimal patient health and minimize patient harm. As part of this larger initiative, a multi-disciplinary team of experts from across the institution convened for 3 years to review the relevant evidence base and develop guidelines appropriate for this institutional context. The goal of the guideline development process was to ensure evidence-based, patient-centered obesity care is available to all patients in our health system that reduces the experience of weight stigma, improves health outcomes, reduces costs, and limits patient harm. As much of weight management starts with provider communication through the use of patient-centered language and motivational interviewing to compassionately engage patients in a conversation about their weight, it was identified that we first should explore patients' experiences of obesity care within our health system prior to identifying areas for improvement [19–21]. The exploratory patient experience information and preferences from this survey will inform a future institutional health system improvement project to promote patient-centered language use and reduce weight stigma, using plan-do-study-act (PDSA) cycles [22].

A subgroup of 7 OHSU interdisciplinary clinicians from the 44-person institutional guideline development team convened in 2020 to explore patient attitudes, behaviors, and institutional factors when receiving prior weight management focused care. The guideline priorities were then brought to OHSU's partnering survey company (NRC Health) for the development of a 15-item survey, including validated questions for demographic data and food insecurity (2-item Hunger Vital Sign), as well as content expert guided questions targeting specific communication and systems issues [23]. The survey had three sections: (1) demographic and healthcare characteristics of patients (supplemented with NRC Health acquired OHSU electronic health record (EHR) data: age, sex, race, marital status, and body mass index or BMI); (2) communication and patient care variables (e.g., weight loss topics discussed, level of comfort bringing up their weight); and (3) preferred communication variables. A five-point scale was used to assess patients' comfort when talking about weight/size (scale: 1 = very uncomfortable; 2 = somewhat uncomfortable; 3 = neither comfortable nor uncomfortable; 4 = somewhat comfortable; 5 = very comfortable). A five-point scale was also used when asking about patients views of communication affinity (scale: 1 = very negative; 2 = somewhat negative; 3 = neither positive nor negative; 4 = somewhat positive; 5 = very positive). The full survey is available as supporting information: **S1 File. NRC OHSU obesity guidelines survey**

Demographic factors, including race, were included as characteristics in this analysis due to the known health inequities that are caused by social, cultural, and structural factors that perpetuate implicit and explicit bias based on individuals' demographics, which may create intersectional challenges for patients experiencing weight and other demographic-related stigma [24, 25].

In addition, three open-ended questions were included to increase the depth of patient responses: (1) "In your own words, what made you uncomfortable when the healthcare provider brought up weight management?"; (2) "In your own words, why do you get uncomfortable bringing up your weight to your healthcare provider?"; and (3) "Is there anything else you would like us to know about healthcare related to weight or size?" Please see the supplemental materials to review the full survey (including question logic) **S1 File. NRC OHSU obesity guidelines survey**. The first two open-ended questions were only asked if patients responded that they were either 'somewhat uncomfortable' or 'very uncomfortable' to the immediately preceding, related single answer multiple-choice questions. After the survey was developed by national survey experts from NRC Health, it was iteratively reviewed by the institutional weight management guideline subcommittee for survey development, as well as the larger multi-stakeholder guideline development team to enhance face and content validity prior to its administration via email between September 29, 2021 and October 5, 2021 [26]. The survey was also translated into Spanish by a certified bilingual healthcare interpreter for primary care clinics that primarily serve Spanish-speaking patients and distributed online through specific clinic partnerships. All project activities were reviewed by OHSU's IRB (STUDYID 00027014), which determined this effort was not human subjects research due to its focus on health system improvement. This manuscript was drafted in adherence with the strengthening the reporting of observational studies in epidemiology (STROBE) and standards for reporting qualitative research (SRQR) guidelines [27, 28].

## Study population & survey administration

The survey population included 16,758 current and former patients seen at OHSU prior to survey fielding. These patients previously electronically opted-in to provide survey feedback after a patient visit to OHSU, becoming a member of NRC Health's (OHSU's survey vendor) established Community Insights Program. These patients received an e-mail with a link to the

survey which was hosted on NRC Health's online platform (see **S1 File** for additional details). Some parents of patients under 18 were included in the Community Insights Program and were kept in the analyses given the exploratory nature of this survey and the overall guideline development focus upon both pediatric and adult obesity.

## Data analyses

For quantitative analyses, descriptive statistics, including frequencies, means, medians, standard deviations, ranges, and percentiles were calculated to assess the shape of the data. Due to the exploratory nature of these analyses, sample size calculations were not performed a priori. Data were sub-grouped by EHR defined sex and by weight categories per BMI: underweight (BMI<18.5 kg/m$^2$), normal weight (BMI 18.5–24.9 kg/m$^2$), overweight (BMI 25–29.9 kg/m$^2$), and obesity ($\geq$ 30 kg/m$^2$). Patient comfort and communication variables were categorized by these weight categories: overweight (BMI 25–29.9 kg/m$^2$), class 1 obesity (30–34.9 kg/m$^2$), class 2 obesity (35–39.9 kg/m$^2$), and class 3 obesity ($\geq$40 kg/m$^2$). Though BMI has limitations in the assessment of obesity and should be coupled with other assessment tools (such as the Edmonton Obesity Staging System) when able, it is the only widely-used measure to characterize obesity in this population and has strong population level associations with comorbidities and mortality [29–32]. Chi-squared was used to assess categorical variables and analysis of variance (ANOVA) was used to assess continuous variables. All assumptions for Chi-squared and ANOVA tests were assessed and met prior to performing any analyses. These statistical tests were chosen based on the presence of categorical and continuous data with the goal to assess overall effects rather than individual interactions across the independent participant survey responses. All tests were two-tailed with alpha levels set at 0.05 for determining statistical significance. Overall missingness was reported in the results or indicated by minimum and maximum sample sizes per question as applicable. We report the sample sizes for each individual question's answer choices by category to display the missingness from each item and denote overall missingness in Table 1. No additional methods were employed to statistically address missingness. Bonferroni corrections were included to account for multiple comparisons.

For qualitative analyses, we used classical content analysis to identify emergent themes within each of the three open-ended survey question responses only among the subgroups of respondents with overweight and class 1–3 obesity [33]. More specifically, authors RMK and PAC reviewed all free text responses using constant comparative analyses techniques to independently identify and then define common themes that emerged from patients' responses. We then used consensus meetings to collapse, split, and finalize the themes' definitions and selected exemplars that reflected the emergent themes. These consensus meetings provided the opportunity to refine and group emergent themes collaboratively and review the free text responses for additional thematic coding. It is important to note that authors RMK and PAC have differing career fields, demographic characteristics, and professional settings, which informed unique interpretations of participant statements to generate iteratively refined thematic codes.

## Results

Of the 16,758 patients in the total sample, 8,714 opened the e-mail (52%), while 3,219 patients completed the survey (response rate = 19.2%) and provided 957 free-text responses for qualitative analysis. Of the 3,219 respondents, 2,816 had available BMI and sex data in our institutional EHR, which were paired to their survey responses for analysis. The average time for survey completion was 6 minutes and 18 seconds. Overall, respondents were predominantly female (57.9%), between 45 and 74 years of age (66.5%), White (87.4%), married (61.1%), and

**Table 1. Demographic and health care characteristics of survey population.**

| Characteristics | Total* (n = 2,816) 100% | Underweight <18.5kg/m² (n = 64 or 2.3%) | Normal Weight 18.5–24.9 kg/m² (n = 855 or 30.4%) | Overweight 25.0–29.9 kg/m² (n = 887 or 31.5%) | Obesity ≥30.0 kg/m² (n = 1,010 or 35.9%) | p-value |
|---|---|---|---|---|---|---|
| **Sex** | | | | | | |
| Female | 1630 (57.9%) | 45 (70.3%) | 545 (63.7%) | 438 (49.4%) | 602 (59.6%) | **<0.001** |
| Male | 1183 (42%) | 19 (29.7%) | 308 (36.0%) | 449 (50.6%) | 407 (40.3%) | |
| *Missing* | 3 (<1%) | 0 (0%) | 2 (<1%) | 0 (0%) | 1 (<1%) | |
| **Age in Years** | | | | | | |
| < 18 | 58 (2.1%) | 34 (53.1%) | 14 (1.6%) | 5 (<1%) | 5 (<1%) | **<0.001** |
| 18–44 | 345 (12.3%) | 6 (9.4%) | 111 (13.0%) | 99 (11.2%) | 129 (12.8%) | |
| 45–64 | 890 (31.6%) | 5 (7.8%) | 224 (26.2%) | 263 (29.7%) | 398 (39.4%) | |
| 65–74 | 983 (34.9%) | 13 (20.3%) | 311 (36.4%) | 319 (36%) | 340 (33.7%) | |
| ≥ 75 | 540 (19.2%) | 6 (9.4%) | 195 (22.8%) | 201 (22.7%) | 138 (13.7%) | |
| **Race** | | | | | | |
| White | 2462 (87.4%) | 50 (78.1%) | 742 (86.8%) | 786 (88.6%) | 884 (87.5%) | **0.004** |
| Non-White | 232 (6%) | 11 (17.2%) | 76 (8.9%) | 60 (6.8%) | 85 (8.4%) | |
| Unknown | 122 (4.3%) | 3 (4.7%) | 37 (4.3%) | 41 (4.6%) | 41 (4.1%) | |
| **Marital status** | | | | | | |
| Single | 1092 (38.8%) | 52 (81.3%) | 323 (37.8%) | 294 (33.1%) | 423 (41.9%) | **<0.001** |
| Married | 1721 (61.1%) | 12 (18.8%) | 530 (62.0%) | 593 (66.9%) | 586 (58.0%) | |
| Unknown | 3 (<1%) | 0 (0%) | 2 (<1%) | 0 (0%) | 1 (<1%) | |
| **Have PCP they visit regularly** | | | | | | |
| Yes, non-OHSU | 1323 (47.0%) | 33 (51.6%) | 378 (44.2%) | 411 (46.3%) | 501 (49.6%) | 0.155 |
| Yes, with OHSU | 1284 (45.6%) | 24 (37.5%) | 406 (47.5%) | 416 (46.9%) | 438 (43.4%) | |
| No | 174 (6.2%) | 7 (10.9%) | 61 (7.1%) | 51 (5.7%) | 55 (5.4%) | |
| Unsure | 35 (1.2%) | 0 (0%) | 10 (1.2%) | 9 (1.0%) | 16 (1.6%) | |
| **PCP has talked with patient about weight management** | | | | | | |
| Yes | 950 (33%) | 9 (14%) | 75 (9%) | 232 (26%) | 634 (63%) | **<0.001** |
| No | 1820 (64.6%) | 53 (82.8%) | 777 (90.0%) | 645 (72.7%) | 345 (34.2%) | |
| Prefer not to answer | 53 (1.9%) | 2 (3.1%) | 4 (<1%) | 16 (1.8%) | 31 (3.1%) | |
| **Patient health support person†** | | | | | | |
| Partner/Spouse | 1520 (54.0%) | 19 (29.7%) | 466 (54.5%) | 519 (58.5%) | 516 (51.0%) | **<0.001** |
| Healthcare Provider | 1227 (43.6%) | 2 (31.3%) | 401 (46.9%) | 408 (46.0%) | 398 (39.4%) | **<0.001** |
| Friends | 733 (26.0%) | 12 (18.8%) | 229 (26.8%) | 212 (23/9%) | 280 (27.7%) | 0.132 |
| Children | 709 (25.2%) | 11 (17.2%) | 237 (27.7%) | 211 (23.8%) | 250 (24.8%) | 0.107 |
| Siblings | 303 (10.8%) | 7 (10.9%) | 99 (11.6%) | 90 (10.1%) | 107 (10.6%) | 0.807 |
| Parents | 204 (7.2%) | 24 (37.5%) | 62 (7.3%) | 53 (6.0%) | 65 (6.4%) | **<0.001** |
| Church/religious community | 89 (3.2%) | 2 (3.1%) | 28 (3.3%) | 28 (3.2%) | 31 (3%.1) | 0.996 |
| Other | 178 (6.3%) | 7 (10.9%) | 48 (5.6%) | 55 (6.2%) | 68 (6.7%) | 0.343 |
| Not needed | 578 (20.5%) | 11 (17.2%) | 175 (20.5%) | 170 (19.2%) | 222 (22.0%) | 0.431 |
| **Household food insecurity screening** | | | | | | |
| Screened + [a] | 345 (12.5%) | 10 (16.1%) | 75 (9.0%) | 74 (8.5%) | 186 (18.8%) | **<0.001** |
| Screened - [b] | 2410 (87.5%) | 52 (83.9%) | 760 (91.0%) | 795 (91.5%) | 803 (81.2%) | |
| Refused Screening | 65 (2.3%) | 2 (3.1%) | 21 (2.5%) | 18 (2.0%) | 24 (2.4%) | |

*Numbers change due to missingness–overall missingness is 1.3%

†Categories not mutually exclusive

[a] Scored as answering either one or both of the food insecurity questions as 'often true' or 'sometimes true.'

[b] Scored as answering both of the food insecurity questions as 'never true.'

**Table 2. Communication and care patterns between provider and patient according to overweight class and patient sex.**

| Communication and Patient Care Variables | Total Overweight & Obesity (n = 1,881) 100% | Overweight 25–29.9 kg/m² (n = 887) 47.2% | Class 1 30–34.9 kg/m² (n = 536) 28.5% | Class 2 35–39.9 kg/m² (n = 226) 12.0% | Class 3 ≥40 kg/m² (n = 232) 12.3% | p-value |
|---|---|---|---|---|---|---|
| **Weight loss topics discussed** | | | | | | |
| *Females** | n (%) (n = 1,031) | n (%) (n = 438) | n (%) (n = 294) | n (%) (n = 135) | n (%) (n = 164) | |
| Physical Activity | 363 (35.2%) | 93 (21.2%) | 103 (34.3%) | 68 (50.4%) | 104 (63.4%) | <**0.001** |
| Eating Patterns | 250 (24.2%) | 51 (11.6%) | 71 (24.1%) | 50 (37.0%) | 78 (47.6%) | <**0.001** |
| Social Factors | 120 (11.6%) | 23 (5.3%) | 30 (10.2%) | 22 (16.3%) | 45 (27.4%) | <**0.001** |
| *Males** | n (%) (n = 850) | n (%) (n = 449) | n (%) (n = 242) | n (%) (n = 91) | n (%) (n = 68) | |
| Physical Activity | 315 (37.1%) | 95 (21.2%) | 108 (44.6%) | 59 (64.8%) | 53 (77.9%) | <**0.001** |
| Eating Patterns | 200 (23.5%) | 53 (11.8%) | 72 (29.8%) | 36 (39.6%) | 39 (57.4%) | <**0.001** |
| Social Factors | 105 (12.4%) | 24 (5.3%) | 37 (15.3%) | 24 (26.4%) | 20 (29.4%) | <**0.001** |
| **Patient's level of comfort bringing up their own weight†** | | | | | | |
| | Mean (SD) | Mean (SD) | Mean (SD) | Mean (SD) | Mean (SD) | |
| *Females* | (n = 1,031) 4.15 (1.15) | (n = 438) 4.41 (1.0) | (n = 294) 4.06 (1.14) | (n = 135) 3.94 (1.27) | (n = 164) 3.77 (1.30) | <**0.001** |
| *Males* | (n = 850) 4.49 (0.92) | (n = 449) 4.58 (0.84) | (n = 242) 4.41 (0.97) | (n = 91) 4.46 (0.97) | (n = 68) 4.25 (1.15) | **0.016** |
| **Patient care delayed or cancelled because of treatment experiences related to weight** | | | | | | |
| *Females* | n (%) (n = 1,031) | n (%) (n = 438) | n (%) (n = 294) | n (%) (n = 135) | n (%) (n = 164) | |
| Have cancelled care* | 50 (4.8%) | 11 (2.5%) | 10 (3.4%) | 9 (6.7%) | 20 (12.2%) | <**0.001** |
| Have delayed care* | 127 (12.3%) | 22 (5.0%) | 34 (11.6%) | 29 (21.5%) | 42 (25.6%) | <**0.001** |
| No | 862 (83.6%) | 403 (92.0%) | 251 (85.4%) | 98 (72.6%) | 110 (83.6%) | <**0.001** |
| *Males* | n (%) (n = 850) | n (%) (n = 449) | n (%) (n = 242) | n (%) (n = 91) | n (%) (n = 68) | |
| Have cancelled care* | 16 (1.9%) | 5 (1.1%) | 6 (2.5%) | 1 (1.1%) | 4 (5.9%) | **0.044** |
| Have delayed care* | 37 (4.4%) | 13 (2.9%) | 8 (3.3%) | 8 (8.8%) | 8 (11.8%) | **0.001** |
| No | 790 (92.9%) | 427 (95.1%) | 226 (93.4%) | 82 (90.1%) | 55 (80.9%) | <**0.001** |

†Scale: 1 = very uncomfortable; 2 = somewhat uncomfortable; 3 = Neither comfortable nor uncomfortable; 4 = somewhat comfortable; 5 = very comfortable.

Entire range of scale used by participants.

*Categories not mutually exclusive with these multi-answer questions, resulting in percentages that may not total 100% by column.

had a primary care provider (PCP) they visit regularly (92.6%) (**Table 1**). Respondents were distributed across weight classes: underweight (2.3%), normal weight (30.4%), overweight (31.5%), and obesity (35.9%). Having overweight/obesity was associated with sex, age, race, and marital status (**Table 1**). More patients with obesity reported having talked with their primary care physicians about weight management compared to the other weight categories (63% vs. 26% or fewer, P<0.001; **Table 1**). Patients with obesity were also more likely to screen positive for food insecurity than patients with a normal weight (18.8% vs. 9%; P<0.001).

**Table 2** reports on communication and patient care variables according to weight category. Physical activity was discussed most often with patients regardless of weight status—followed by eating patterns and social factors among both female and male patients (p<0.001 for all variables). Although comfort bringing up their weight tended to decrease with increasing weight class for both males and females, this was especially so for females [females with class 3 obesity = 3.77 (SD 1.3) vs. males with class 3 obesity = 4.25 (SD 1.15)]. Having class 3 obesity was

also associated with more cancelled or delayed healthcare visits, especially among females (12.2% of females with class 3 obesity cancelled care and 25.6% delayed care compared to other weight categories; p<0.001, whereas 5.9% of males with class 3 obesity cancelled care compared to other weight categories; p≤0.044 and 11.8% delayed care compared to other weight categories; p = 0.001).

Table 3 shows patients' preferred communication styles according to BMI and sex (obtained from EHR data). Among females, those with overweight and class 1 obesity were more likely to prefer open and direct communication compared to those with classes 2 and 3 obesity (56.6% and 54.1% vs. 41.5% and 45.7%; p = 0.005). The higher the BMI, the greater the preference was for gentler and slower communication. However, direct (52.2%), empathetic (34.6%), and clear (37.5%) communication was generally favored across weight classes for females. Overall, only 2.6% of females reported not ever wanting to talk with their providers about weight management, which was not statistically different according to weight class. Among males, there were no communication differences according to weight class with preferences for direct communication (62.1%) and wanting clarity on details (40.7%). Males (1.3%) rarely indicated they never wanted to talk with their providers about weight management, which did not differ according to weight category. Overall, both male and female respondents felt somewhat positively about the terms: "healthy eating plan," "exercise," "activity," "weight reduction," and "weight loss." Female respondents had statistically significant differences in term of preferences by weight class, with increasing weight class trending towards more neutral or negative responses to all terms.

Qualitative findings from the classical content analysis of open-ended questions are reported in Table 4. Across all three open-ended questions, there were 957 unique responses analyzed by weight class (overweight = 354, class 1 obesity = 280, class 2 obesity = 137, and class 3 obesity = 186). Twenty-seven emergent themes fell into three domains, including: (1) Emotional Hinderances, which we defined as emotive reactions to weight issues that impede progress with weight management; (2) Perceptual Hinderances, defined as perceived or actual experiences that impede progress with weight management; and (3) Perceived Helpfulness, defined as perceived or actual experiences that helped patients with weight management.

The Emotional Hinderance domain had 6 themes that included "Denial" or the perception that weight is not really a problem or recognition that they are denying this, which may affect communication. It also included "Mental Health Effects" or recognition that weight can affect mood and/or is associated with mental illness, including disordered eating. "Discomfort/ Shame and Embarrassment" emerged as a theme, which was a sense that conversations about weight will be awkward or will trigger patient's feelings of humiliation, which leads to avoidance of a conversation. "Beyond One's Control" reflected a sense that weight is not something they can really control; rather, it is due to a co-existing medical condition or treatment or another factor they believe is beyond their control. "Frustration/Guilt/Sadness" reflected feelings of annoyance, self-blame and/or sorrow about the inability to lose weight after many individual attempts or provider recommendations. Lastly, "Self-Awareness/Image Consciousness" reflected recognition that one's current weight status or concerns about the way they look were present without provider discussion.

The domain Perceptual Hinderances contained 7 themes. These themes ranged from components of interpersonal interactions with providers to perceived structural or health system impediments. "Health System Frustration" related to exasperation over health system structure for devaluing prevention, while "Dismissed" expressed perceptions that the patient's concerns were minimized or ignored by their providers. Respondents had concerns over some providers' ability to provide evidenced-based information on the topic of weight management— under the "Distrust" theme. "Measurement Concerns" brought up skepticism over the validity

**Table 3. Patients' preferred communication according to obesity class and sex.**

| Preferred Communication Variables | Total Overweight & Obesity (n = 1,881) 100% | Overweight 25–29.9 kg/m² (n = 887) 47.2% | Class 1 30–34.9 kg/m² (n = 536) 49.8% | Class 2 35–39.9 kg/m² (n = 226) 22.9% | Class 3 ≥40 kg/m² (n = 232) 27.2% | p-value |
|---|---|---|---|---|---|---|
| **How would you want your provider to talk to you about weight management** | **n (%)** | **n (%)** | **n (%)** | **n (%)** | **n (%)** | |
| *Female* | *(n = 1,031)* | *(n = 438)* | *(n = 294)* | *(n = 135)* | *(n = 164)* | |
| Directly (Open & Honest) | 538 (52.2%) | 248 (56.6%) | 159 (54.1%) | 56 (41.5%) | 75 (45.7%) | **0.005** |
| Gently (May be hard for me to hear) | 192 (18.6%) | 68 (15.5%) | 48 (16.3%) | 31 (23.0%) | 45 (27.4%) | **0.003** |
| Slowly (Don't want to feel rushed) | 105 (10.2%) | 40 (9.1%) | 23 (7.8%) | 13 (9.6%) | 29 (17.7%) | **0.006** |
| Empathetically (Want to know you are interested in my feelings) | 357 (34.6%) | 140 (32.0%) | 101 (34.4%) | 46 (34.1%) | 70 (42.7%) | 0.107 |
| Simply (Don't want details) | 82 (8.0%) | 35 (8.0%) | 19 (6.5%) | 10 (7.4%) | 18 (11.0%) | 0.392 |
| Clearly (Want details) | 387 (37.5%) | 173 (39.5%) | 111 (37.8%) | 46 (34.1%) | 57 (34.8%) | 0.582 |
| I don't ever want to talk to providers about weight management | 27 (2.6%) | 9 (2.1%) | 9 (3.1%) | 3 (2.2%) | 6 (3.7%) | 0.671 |
| *Male* | *(n = 850)* | *(n = 449)* | *(n = 242)* | *(n = 91)* | *(n = 68)* | |
| Directly (Open & Honest) | 528 (62.1%) | 273 (60.8%) | 155 (64.0%) | 57 (62.6%) | 43 (63.2%) | 0.859 |
| Gently (May be hard for me to hear) | 82 (9.6%) | 42 (9.4%) | 23 (9.5%) | 9 (9.9%) | 8 (11.8%) | 0.939 |
| Slowly (Don't want to feel rushed) | 53 (6.2%) | 24 (5.3%) | 13 (5.4%) | 9 (9.9%) | 7 (10.3%) | 0.178 |
| Empathetically (Want to know you are interested in my feelings) | 183 (21.5%) | 99 (22.0%) | 45 (18.6%) | 21 (23.1%) | 18 (26.5%) | 0.491 |
| Simply (Don't want details) | 52 (6.1%) | 30 (6.7%) | 12 (5.0%) | 8 (8.8%) | 2 (2.9%) | 0.370 |
| Clearly (Want details) | 346 (40.7%) | 190 (42.3%) | 92 (38.0%) | 40 (44.0%) | 24 (35.3%) | 0.448 |
| I don't ever want to talk to providers about weight management | 11 (1.3%) | 3 (<1%) | 5 (2.1%) | 1 (1.1%) | 2 (2.9%) | 0.691 |
| **How would you feel about your provider's use of these words when talking with you or a loved one about weight** | Mean (SD) | Mean (SD) | Mean (SD) | Mean (SD) | Mean (SD) | |
| *Female* | *(n = 813–1,011)\** | *(n = 345–435)\** | *(n = 240–287)\** | *(n = 108–131)\** | *(n = 118–158)\** | |
| BMI† | 3.74 (1.22) | 3.98 (1.14) | 3.74 (1.27) | 3.54 (1.15) | 3.19 (1.22) | **<0.001** |
| Weight loss† | 4.03 (1.00) | 4.12 (0.98) | 4.04 (1.03) | 3.90 (1.03) | 3.83 (0.99) | **0.011** |
| Overweight† | 3.79 (1.14) | 3.92 (1.12) | 3.87 (1.15) | 3.54 (1.18) | 3.50 (1.06) | **<0.001** |
| Obesity† | 3.31 (1.46) | 3.51 (1.41) | 3.35 (1.48) | 3.07 (1.42) | 2.83 (1.45) | **<0.001** |
| Diet† | 3.97 (1.13) | 4.16 (1.05) | 3.99 (1.13) | 3.77 (1.16) | 3.52 (1.22) | **<0.001** |
| Healthy eating plan† | 4.36 (0.88) | 4.50 (0.79) | 4.42 (0.83) | 4.11 (0.99) | 4.06 (1.03) | **<0.001** |
| Weight gain† | 3.79 (1.08) | 3.94 (1.07) | 3.80 (1.10) | 3.71 (1.04) | 3.40 (1.05) | **<0.001** |
| Weight reduction† | 4.08 (0.98) | 4.19 (0.98) | 4.13 (0.98) | 3.85 (1.00) | 3.85 (0.90) | **<0.001** |
| Exercise† | 4.25 (0.91) | 4.41 (0.85) | 4.32 (0.91) | 4.00 (0.95) | 3.93 (0.92) | **<0.001** |
| Activity† | 4.30 (0.86) | 4.43 (0.85) | 4.36 (0.86) | 4.06 (0.93) | 4.03 (0.89) | **<0.001** |
| *Male* | *(n = 772–836)\** | *(n = 300–445)\** | *(n = 205–237)\** | *(n = 83–90)\** | *(n = 57–67)\** | |
| BMI† | 4.08 (1.03) | 4.16 (0.96) | 4.01 (1.06) | 3.92 (1.20) | 3.97 (1.06) | 0.080 |
| Weight loss† | 4.30 (0.86) | 4.32 (0.87) | 4.26 (0.85) | 4.44 (0.80) | 4.15 (0.96) | 0.188 |
| Overweight† | 4.15 (0.98) | 4.21 (0.95) | 4.09 (0.96) | 4.02 (1.06) | 4.03 (1.05) | 0.172 |
| Obesity† | 3.91 (1.19) | 4.00 (1.12) | 3.83 (1.27) | 3.76 (1.27) | 3.86 (1.22) | 0.209 |
| Diet† | 4.28 (0.92) | 4.33 (0.89) | 4.27 (0.90) | 4.16 (1.09) | 4.19 (0.97) | 0.317 |
| Healthy eating plan† | 4.48 (0.80) | 4.53 (0.77) | 4.43 (0.84) | 4.49 (0.77) | 4.32 (0.94) | 0.146 |
| Weight gain† | 4.07 (0.97) | 4.12 (0.94) | 4.04 (0.96) | 3.93 (1.09) | 4.00 (1.02) | 0.317 |
| Weight reduction† | 4.29 (0.88) | 4.34 (0.87) | 4.22 (0.85) | 4.29 (0.98) | 4.18 (0.99) | 0.282 |
| Exercise† | 4.46 (0.81) | 4.56 (0.72) | 4.37 (0.86) | 4.42 (0.90) | 4.13 (0.97) | **<0.001** |

(*Continued*)

**Table 3.** (Continued)

| Preferred Communication Variables | Total Overweight & Obesity (n = 1,881) 100% | Overweight 25–29.9 kg/m² (n = 887) 47.2% | Class 1 30–34.9 kg/m² (n = 536) 49.8% | Class 2 35–39.9 kg/m² (n = 226) 22.9% | Class 3 ≥40 kg/m² (n = 232) 27.2% | p-value |
|---|---|---|---|---|---|---|
| **How would you want your provider to talk to you about weight management** | **n (%)** | **n (%)** | **n (%)** | **n (%)** | **n (%)** | |
| Activity† | 4.47 (0.79) | 4.56 (0.74) | 4.44 (0.78) | 4.41 (0.83) | 4.06 (0.94) | **<0.001** |

†Scale: 1 = very negative; 2 = somewhat negative; 3 = neither positive nor negative; 4 = somewhat positive; 5 = very positive. Full scale range was used by participants

*The sample size (n) listed for each column (sub-grouped by sex) is listed as the minimum and maximum for ease of interpretation, as there was variability in participant response to each term.

of weight classifications (e.g., BMI). "Unrelatable Providers or Role Models" was defined as resistance to provider recommendations or perceived reliability of their recommendations based on the provider's appearance or emphasis. Where "Perceived Mismatch" highlighted mismatch between provider concerns about obesity and patient satisfaction with or dis-interest in discussing their weight. Lastly, "Misinformation" described concerns that societal perceptions went against medical recommendations for health and wellness.

The third domain of "Perceived Helpfulness" was made up of 14 themes. "Empathetic Communication" (the recognition of the need for a compassionate approach, shared decision making, and unbiased conversation) and "Weight in Context of Wholistic Health Care" (perception that weight management needs to be considered within the patient's overall health circumstances) were two of the most prevalently discussed themes. "Patient Centered Care Plan" noted that patients desire providers to develop a health plan jointly with them. "Relationship Building" with the care team to address weight management and "Community Support" from family, friends, and/or the community were both expressed by respondents. Several respondents indicated the desire for "Direct Communication" from providers or noted the need for additional emphasis on "Eating Patterns" or "Exercise." Other respondents were interested in an "Intensive Program or Subspecialist Care," "Medication Assisted Treatment," "Technological Supports," or "More Information" (regarding patient-centered lifestyle changes or therapeutic options). Respondents indicated importance on the "Clinical Environment," regarding clinic workflows, the hospital environment, and the physical environment/furniture being built to encourage health and accommodate people of all sizes. Other respondents noted "Satisfaction" or feeling positively about their current weight status.

## Discussion

Respondents to this survey noted distinct preferences in the way the conversation is approached, and the language providers used, according to both BMI and sex (male and female). Very few individuals with obesity (1–3%) did not want to discuss weight management at all with their providers. In contrast, a high number of respondents (73% of those with overweight and 34% of those with obesity) indicated they had never discussed weight management with their providers. Previous research suggests that patients and providers may not always agree on whether weight management has been discussed during visits [34]. Despite this, it is strongly recommended for providers, with permission, to increasingly include the topic of weight management in their care and to be explicit about it, especially given the impact brief discussions can have upon patient weight loss [9, 35]. This is also in the context of personal healthcare providers serving as the most heavily relied upon and second most-trusted source

**Table 4.** Qualitative findings according to primary thematic domains, emergent themes, descriptions, and reflective exemplars.

| Emergent Theme | Description | Exemplars |
|---|---|---|
| *Emotional Hinderances–Emotive reactions to weight issues that impede progress with weight management* | | |
| Mental Health Effects | Recognition that weight can affect mood and/or is associated mental illness, including disordered eating. | "Take into consideration the reasons WHY someone is overweight like health, binging problems, or depression/loneliness" *[Participant #OC1†-242]*<br>"I felt like HE REALLY DID NOT UNDERSTAND that losing weight is not only changing certain eating habits, some people need some emotional & mental health support." *[Participant #OC3†-220]* |
| Beyond One's Control or Illness Associated | Sense that weight is not something they can really control. Rather, it is due to a co-existing medical condition or treatment or another factor they believe is beyond their control. | "It's something I've successfully dealt with through exercise until my Lupron treatment." *[Participant #OC1-84]*<br>"I feel I have little control of my weight because of medical conditions so am embarrassed when I have to weigh in." *[Participant #OC2†-15]* |
| Denial | Perception that weight is not really a problem or recognition that they are denying this, which may affect communication. | "I'm not worried at all about my weight. . ." *[Participant #OW†-221]*<br>"My PCP has not brought up weight management with me. . .It isn't a concern of mine." *[Participant #OC1-294]* |
| Discomfort/ Shame/ Embarrassment | Sense that conversations about weight will be awkward or will trigger patient's feelings of humiliation which leads to avoidance of a conversation. | "I know that I am overweight and don't need to be shamed about it." *[Participant #OW-31]*<br>"Don't like to be reminded of my failure to improve." *[Participant #OC3-68]* |
| Frustration/ Guilt/ Sadness | Feelings of annoyance, self-blame and/or sorrow about inability to lose weight after many individual attempts or provider recommendations. | "Because it's my own fault and I have not been successful in losing or keeping weight off." *[Participant #OC2-46]*<br>"I appreciate my Providers attitude and empathy towards my situation. I have a very low self- esteem from being abused as a child even to this day I feel like everything is my fault." *[Participant #OC3-124]* |
| Self-awareness/ Image Consciousness | Recognition of one's current weight status or concerns about the way they look without provider discussion. | "It is just a hard subject it's been an issue I have fought with for my entire life" *[Participant #OC2-51]*<br>"It is easy to speak about weight with my primary care provider because of mirrors. I can see myself in a mirror and my inner voice will say '. . ., you're fat.'" *[Participant #OC3-163]* |
| *Perceptual Hinderances–Perceived or actual experiences that impede progress with weight management* | | |
| Health System Frustration | Respondents expressing exasperation over health system structure for devaluing prevention through costs of care and perceived emphasis. | "Doctors get way too little education about nutrition and the American system of responding only to crises combined with how healthcare providers are paid [for heroic actions rather than overall good outcomes and prevention] make [it so] that patients better self-educate, advocate and, above all, act." *[Participant #OW-262]*<br>"I was unable to get into a bariatric clinic with OHSU, I opted to go to Mexico to get the gastric sleeve" *[Participant #OC3-147]* |
| Dismissed | Perceptions that the patient's concerns were minimized or ignored by their providers. | "I would love a doctor to actively help me manage that issue. I think since it isn't affecting my bloodwork they just don't consider it a problem." *[Participant #OW-506]*<br>"I think that medical professionals will often cut people off about other medical concerns and blame everything on being overweight. Patients need to have their other concerns taken care of and not being ignored while their weight is the main focus" *[Participant #OC1-94]* |
| Distrust | Lack of confidence in their healthcare provider's ability to provide accurate, evidence-based information on the topic of weight management. | "Judgement, antiquated ideas, medicine has zero help to provide. I have no trust because doctors have almost no training in this area." *[Participant #OW-650]*<br>"I don't believe in BMI. I don't trust most doctors with nutritional advice. I don't believe I gastric surgery for weight management." *[Participant #OC3-232]* |
| Measurement Concerns | Skepticism regarding the way weight is classified (e.g., often questioning the validity of using Body Mass Index). | "BMI is nonsense for individual weight assessment and should be discarded by health care professionals entirely. Before there is any discussion of weight loss, there should be a clear reason for it. . ." *[Participant #OC1-216]*<br>"I have a morbidly obese BMI, but I am athletic, play several sports, and can run 3 miles and hike about 7 miles. I'm tired of OHSU using BMI still. BMI needs to go away. Yes, my PCP has talked about my weight but I don't think she is worried about it. BMI is archaic and I can't believe it's still used today." *[Participant #OC2-36]* |

*(Continued)*

**Table 4.** (Continued)

| Emergent Theme | Description | Exemplars |
|---|---|---|
| Unrelatable Providers or Role Models | Resistance to provider recommendations or perceived reliability of their recommendations based on the provider's appearance or emphasis. | "There have been times I have avoided appointments because of my weight or weight gain. It is hard because so many young Physicians are thin and have never had any issues related to wait and no one in their family has and they think all people who are overweight are lazy. . . They don't understand that we have" [Participant #OC3-169]<br><br>"Most of my discomfort was due to the doctor was vegan or vegetarian. He looked like he had NEVER had a weight problem in his entire life. . ." [Participant #OC3-220] |
| Perceived Mismatch | Mismatch between provider concerns about obesity and patient satisfaction with or dis-interest in discussing their weight. | "Healthcare provider did a poor job with the discussion and it was clear how they feel very negatively or look down on people who have obesity." [Participant #OC3-90]<br><br>"Previous bad experiences with other doctors. I still worry that seeing my doctor is contingent on my weight status, or receiving care for other issues. Often times I put off doctor visits when I am not losing enough weight, which concerns me that I may have other issues developing that I just may not know about. I anticipate judgment, or being sent out with simply advice to lose weight to solve my issue (whatever it might be)." [Participant #OC3-101] |
| Misinformation | Concerns over societal perceptions regarding health and wellness that go against medical recommendations. | "For dietary considerations, it seems like there is tons of conflicting information about what is actually healthy for both good nutrition and weight loss." [Participant #OC2-171]<br><br>"There are so many diet fads, it's difficult to know which one is for you." [Participant #OC3-40] |
| *Perceived Helpfulness—Perceived or actual experiences that helped patients with weight management* | | |
| Empathetic Communication | Recognition of the need for a compassionate approach, shared decision making, and unbiased conversation when discussing weight status. | "I would like providers to be versed in the language/concepts/ thinking around weight shaming and be able to acknowledge that philosophy in the way they talk about health" [Participant #OC1-149]<br><br>"My preference is for providers to approach in a friendly empathetic way with a gradual supportive approach. Some providers have an approach which is demeaning and ineffective in my opinion. . ." [Participant #OC2-212] |
| Weight in Context of Wholistic Health Care | Perception that weight management needs to be considered within the patient's overall health circumstances, including their social determinants of health and often espousing a health at every size mentality. | "It's hard to work full-time, do family activities, meal plan, be sick, and take care of myself. I either need superpowers. Or more time (which usually means less than 5 hours sleep)" [Participant #OC2-141]<br><br>"The under use of the concept fit at any size. Weight is one of the hardest issues to discuss with a health provider because of its negative connotations and the many weight-related biases." [Participant #OC3-113] |
| Patient Centered Care Plan | Patient's desire for providers to develop a health plan jointly with them, schedule regular follow up, and encourage progress towards attainable goals. | ". . .So you walk out with Dr. telling u need to lose weight, another brick on your back to deal with and no solution, or help as to how to get out from under. Yeah, I have put off visits because of this." [Participant #OW-531]<br><br>"In general, would like the medical profession to start with healthy things such as diet and exercise first and work out a plan and then re-evaluate weight and if for whatever reason [the] pt. can't do [it] then suggest other possible solutions such as meds or surgery. . ." [Participant #OC3-67] |
| Relationship Building | Perceived importance placed on having a supportive relationship with the care team to address weight management. | ". . .I quit trying to communicate about any of it because she doesn't bother to refresh herself on my condition or what's going on in my life." [Participant #OC2-61]<br><br>"I am lucky that I have an amazing PCP right now so feel comfortable with any conversations. That has not always been true with other healthcare providers. Bottom line for me is not necessarily the words or conversation as it is the relationship and trust I have with my PCP." [Participant #OC3-187] |
| Community Support | Recognition of the importance of family, friends, and/or the community upon healthy behaviors and weight. | "My spouse is overwhelmingly supportive of a healthy diet, exercise and living a good life." [Participant #OC1-56]<br><br>"I appreciated when my physician said to me that you could come off the amount of medication if I lost weight. My family, friends, and a very close physician friend also expressed concern about my weight. Although my family didn't use the term obesity. I came to realize that I was obese and had a BMI of 44. I have lost 64 pounds since April 2021. I have a nutritionist/dietitian. I had knee replacement surgery and I am healing nicely." [Participant #OC3-18] |

*(Continued)*

**Table 4.** (Continued)

| Emergent Theme | Description | Exemplars |
|---|---|---|
| Direct Communication | Preferences for a direct approach to discussing weight management with providers. | "I am obese, but, if anything, healthcare workers that I interact with seldom mention it or try to tread lightly about it. I'd prefer them to speak plainly and directly about it." *[Participant #OC2-124]*<br>"Matter of fact discussion is my preference. It's not like I don't know I'm fat—I do. However, discrimination because of weight is a \*real\* problem. A healthcare professional should \*never\* be mean or cruel..." *[Participant #OC3-38]* |
| Eating Patterns | Importance placed on the impact of dietary patterns on weight status. | "Ask your patients what they eat! Encourage them to make incremental adjustments to their diets." *[Participant #OW-646]*<br>"I think dieting is harmful to your health. Finding eating habits or establishing eating habits that are healthy by learning about nutrition is more successful than diets, also finding exercise routines that are enjoyable are important. I have lost about 60 pounds, and kept it off for about a year so far, I would like to lose a little weight but I'm not really changing my good eating habits to do it, I'm just spending more time at the gym, which I personally really enjoy." *[Participant #OC1-190]* |
| Exercise | Patients' desire more emphasis on the importance of physical activity on weight status. | "I've battled weight all my life and know the secret for me is exercise." *[Participant #OC2-20]*<br>"I would like someone to help with setting up an exercise program that is a fit for me and I think most people that are overweight would like that. Also, to have the cost be within a person's budget." *[Participant #OC3-168]* |
| Intensive Program or Subspecialist Care | Perceived importance of discussion of subspecialist or team-based care (surgical, dietician, culinary medicine program, or other sub-specialist referral). | "Weight loss surgery was the best thing I have ever done for myself I am so grateful for the staff and my Dr. for supporting me!" *[Participant #OC1-8]*<br>"...A true weight loss program including support, medical treatment when necessary and medications would be most valuable. Active participation helps." *[Participant #OC1-220]* |
| Medication Assisted Treatment | Patient interest or inquiry into the use of anti-obesity medications. | "Med assisted weight loss is realistic. After education and practice managin' diet. Meds before surgery. Education before meds." *[Participant #OC2-190]*<br>"I really can't be trusted to not over eat! It's all my own fault! I wish there was a medicine I could take to stop my appetite! Like in the 80's there were these pills called black beauties they were awesome for that!" *[Participant #OC3-172]* |
| Technologic Supports | Patients prefer using online or App based programs to help manage their weight. | "I would love to see a supportive/interactive on-line weight loss program that provides meal plans based on food likes, times of day to eat, medical goals or conditions, etc..." *[Participant #OC1-335]*<br>"Maintenance Phase podcast is an amazing resource that is a local Portland host. The podcast goes into history of various health topics and common myths regarding weight loss..." *[Participant #OC3-205]* |
| More Information | Patients wanting more patient-centered information regarding lifestyle changes, co-morbid conditions, and therapeutic options. | "I like to have all the info, good or bad and especially bad as it help make a better lifestyle choice" *[Participant #OW-200]*<br>"I would appreciate open conversation information and support on health care related to my weight. For some of us healthy foods are not affordable" *[Participant #OC1-255]* |
| Clinical Environment | Clinic workflows, hospital environment, and physical equipment/furniture should be built to encourage health and accommodate people of all sizes. | "I also think the medical profession needs to back up the healthy eating they recommend by providing it overwhelmingly in hospitals and clinics, whether it's serving it to patients or offering it as an option, as in a cafe. I've never been impressed with the healthy eating options when I've been hospitalized anywhere, and have struggled to find good options when I've been in a waiting room or such for any length of time." *[Participant #OW-601]*<br>"Have chairs/furniture/exam gowns sized for larger people. It's humiliating to wear a gown that won't close all the way or to sit in a chair that we have to squeeze into and/or worry about breaking." *[Participant #OC3-203]* |
| Satisfaction | Feeling positively about their current weight status. | "Feeling in a good place at present with my weight." *[Participant #OW-493]*<br>"I like my weight and have a vigorous life." *[Participant #OW-676]* |

†Participant weight class notation is as follows: OW = overweight per BMI 25–29.9 kg/m$^2$ (n = 887, 47.2%), OC1 = obesity class 1 per BMI 30–34.9 kg/m$^2$ (n = 536, 28.5%), OC2 = obesity class 2 per BMI 35–39.9 kg/m$^2$ (n = 226, 12.0%), OC3 = obesity class 3 per BMI ≥40 kg/m$^2$ (n = 232, 12.3%). Thematic exemplars displayed in aggregate per participant weight class from n = 957 total responses to three individually analyzed open-ended questions: (1) "In your own words, what made you uncomfortable when the healthcare provider brought up weight management?"; (2) "In your own words, why do you get uncomfortable bringing up your weight to your healthcare provider?"; and (3) "Is there anything else you would like us to know about healthcare related to weight or size?"

(behind registered dietician nutritionists) of nutrition information [36]. That said, Table 2 demonstrates that patients less often recall providers discussing factors beyond physical activity, which plays a relatively minor role in weight management as compared to eating patterns and social factors. Future work should evaluate both the patient and provider perspective of weight related conversations in near real time to better assess this finding.

Qualitative findings indicated that when some patients presented to appointments with seemingly unrelated concerns, it was stigmatizing and off-putting for providers to elevate the topic of weight management (see Table 3 *Participant #OC3-101* quotation from the "Perceived Mismatch" theme). In addition to being specific about when weight management is being addressed, including information about why weight management is related to presenting concerns may aid in reducing patients' feelings of being stigmatized. Related to the recommendation to increase discussions of weight management in primary care, physician training is needed to increase both confidence and sensitivity in effective communication methods [9, 37]. Several individuals mentioned the desire for a greater focus on health at every size (see the "Weight in Context of Wholistic Health Care" theme). This perspective can refocus the patient-provider conversation towards a weight-neutral approach, encouraging health promotion for patients of all body sizes that may be more responsive to an individual patient's needs [38].

There is no singular approach to addressing weight management with all patients, given their individualized preferences and perspectives [9, 39]. One recommendation gleaned from participants' free responses and the variability in patient preferences is for providers to broach the discussion of weight management and lifestyle changes by first asking patients for their permission to do so. This is consistent with the themes of "Patient Centered Care" and "Relationship Building" that arose in the qualitative data, in addition to guidelines published by Gallagher and colleagues in 2021 [40]. Our findings (across all BMI categories) indicate that respondents preferred direct, clear, and empathetic styles of communication. This is similar to Koball, et al. who reported that patients preferred "very direct/straightforward" communication (see Table 3 "Direct Communication" theme) [41]. However, females with class 2 and 3 obesity less often preferred a direct approach (see Table 3 *Participant #OC2-212* "Empathetic Communication" theme). We additionally found that females with higher a weight class were more likely to prefer gentler and slower communication styles. Koball, et al. found a similar pattern of preferences: patients in higher BMI categories wanted providers to discuss weight sensitively (though Koball's study did not stratify by sex) [41].

The sex differences in the quantitative results were particularly pronounced. In contrast to female patients, male patients had no statistically significant differences in how they would want a provider to discuss weight management. Additionally, female individuals had statistically significant differences in their affinity for specific terms, with all terms being less favorable with increasing weight class among females. Similar to the approach, males had relatively few significant differences in term preferences across weight classes, though exercise and activity were seen as less favorable among males with class 3 obesity. The most concerning of these findings was the relatively high rates of delayed or canceled care for female respondents, reaching 25.6% and 12.2% for females with class 3 obesity. Males also had increasing rates of delayed or canceled care, though this only reached 11.8% and 5.9% for males with class 3 obesity. Relatively few studies have had such a robust sample of male respondents. These sex differences may be driven by differential experiences of both implicit and explicit weight bias that leads to higher rates of weight-bias internalization among female patients as compared to male patients. Weight bias internalization occurs when an individual applies a negative weight-related stereotype to themselves that fosters a narrative of self-derogation [42]. The qualitative results for the themes "Discomfort/ Shame/ Embarrassment" and "Frustration/ Guilt/ Sadness" displayed features of self-blame and adoption of negative self-opinions driven by prior

experiences. Similar to our findings, a recent international study of weight bias internalization found independent associations with healthcare avoidance, perceived judgment by health providers, worse patient-provider communication, and lower healthcare quality [13].

As awareness increases about the harms of weight bias perpetuated by healthcare providers and systems, identifying language and approaches that are least likely to harm patients is critical for healthcare systems to move forward with weight management initiatives. Our finding of lower patient preference for the term "obesity" is similar to prior findings as summarized in a recent systematic review [43]. Puhl's systematic review reported that phrases incorporating the term "BMI" or "weight" were most preferred and those including the word "obese" or informal terms such as "heavy," "chubby," and "fat" were consistently least preferred across studies among community, primary care, and treatment-seeking populations [43]. In our sample, preferences for terms varied by weight category for females (all words were rated lower by females in higher compared to lower BMI categories) without significant variation for males. In an online community sample, Lydecker, et al. found a similar trend with participants with overweight or obesity rating terms "BMI" and "unhealthy BMI" as less desirable compared to those with a normal weight [44]. In contrast, Puhl, et al. found that patients with obesity showed higher preference for neutrally rated terms "high BMI" and "overweight" compared to normal weight respondents [45]. Neither of these studies evaluated preference trends by BMI and sex, whereas our data showed statistically significant differences by weight class and sex [44, 45].

Strengths of our data are that we included a relatively large health system-specific population not restricted to patients seeking weight management and applicable for informing a health systems approach to screening and management. The large sample size allowed us to examine survey data for differences in experience and preference stratified by EHR defined sex and BMI category. We assessed preferences for terms describing weight management modalities (e.g., "healthy eating plan" and "physical activity") and preferred communication style on which fewer data are available in the literature. Additionally, we included qualitative data to provide a richer understanding of emotional and perceived hinderances as well as perceived helpfulness that can more deeply inform strategies to educate providers on how to initiate discussions about weight across a spectrum of patients in a single healthcare system with an aim to do no harm.

There are several limitations of this analysis, which focused on the adult patient population served by our single institution. Although the single institution focus was the intent, generalizations should be made with caution, and geographic and cultural differences should be accounted for within any individual healthcare system. Future multi-institution surveys across a range of geographic locations would enhance the generalizability of this information, though would be less specific to the individual health system context. In addition, there was a moderate response rate at 19.2% with a possibility of response bias towards those with personal experience of weight stigma, or in contrast, those who experienced less stigma and were more comfortable responding. The possibility of response bias here may be particularly pronounced when considering term and approach preferences by weight class and should be kept in mind when considering these results. Moreover, respondents to the open-ended questions may be more susceptible to response bias based on prior experiences. For these reasons, we sought a mixed methods approach to data collection and analysis. Individual question item non-response may accentuate the potential impact upon the generalizability of these findings, though overall missingness was low and transparently displayed for each item through the displayed sample sizes in each table. The opt-in nature of this online survey may similarly impact non-response bias, as compared to direct surveying of patients as they engaged with care. Relatedly, the cross-sectional nature of this survey provides a single snapshot in time for these

patients and relies on recall of their past experiences. Longitudinally surveying patients, particularly as health system improvements are implemented, would provide more meaningful insights into the direct impact of weight stigma and language use preferences with enhanced confidence in these associations.

There is a slightly higher rate of obesity in the survey population (35.9%) compared with the Oregon population overall (30.4%), although it is more similar to the U.S. prevalence (42.4%) [2]. A primary focus of this survey was to assess patient perspectives in the primary care setting with over 90% of our population regularly following with a primary care provider. However, participant responses were not limited only to the primary care setting and may reflect responses from other health contexts. These participants may also be disproportionately engaged in their health care given the high amount of reported primary care engagement. Follow up studies should focus on examining differences between responders and non-responders to clarify the validity of the data for the broader population served. The overall survey had very few previously validated questions to assess patient preferences, was exploratory in nature, and was not pre-tested among a pilot population prior to full dissemination; though it was developed in partnership with a national third-party vendor and informed by iterative multidisciplinary clinician expert meetings to establish face and content validity. Future partnerships between clinicians and national survey vendors contracted with health systems should consider pilot testing to enhance survey comprehension, clarity, and appropriateness prior to full dissemination. Subsequent iterations of this pilot testing could consider using more nuanced scales to more completely express patient preferences.

We did not include specific term or communication variable definitions, leaving this to individual participant interpretation. This reduced overall survey burden but creates the possibility of error from individual participant interpretations of meaning. That said, the mixed methods analysis of related open-ended and multiple-choice questions with similar qualitative and quantitative findings provide additional confidence in our results. Our focus was to identify variability in patient experiences and preferences by widely recognized weight classifications, as clinicians discuss weight more often among patients with higher BMIs [46]. Future studies may explore patient experiences and language use preferences keeping BMI as a continuous outcome in a regression model to enhance the statistical detail of using BMI over weight class categories. This would also allow for adjustment for underlying comorbidities, which may impact patients' experience of weight stigma.

Importantly, the current data disproportionately represented the views of middle-aged to older adults, females, and White individuals. Despite efforts to include voices from the large Hispanic population in Oregon by translating the survey into Spanish and specifically surveying a Spanish-speaking patient group, response rates were quite low for this group. Thus, results should be interpreted with the understanding that discussion with patients, their language preferences, and their experiences with weight bias may be different for other demographic groups. We have learned from these challenges and hope others will have greater success in future studies with tailored efforts towards inclusive data collection practices, as encouraged by the National Institutes of Health to gather the perspective of patients from various demographic backgrounds [43, 47]. For example, seeking information via phone call, focus group, in person or paper survey, providing multi-modal reminder messages, and including staff with representative backgrounds for direct contact may be more likely to elicit responses from some patients than an online survey. Additionally, including representative patients in project method and survey development is an increasing and important practice for improving access to representative data. All of these methods to enhance the diversity of the sample, in addition to the provision of compensation for survey completion, would likely have the added benefit of enhancing survey response rates, mitigating the potential impact of

non-response bias. Lastly, it is important that we as clinicians, researchers, and healthcare teams critically assess our own language use, keep up with evolving language use preferences, and work towards adopting terms that promote inclusivity and reduce all forms of stigma. This includes our use of race as a sole identifier, which could have been more inclusive by using patient-reported ethnicity.

This initiative sought to use guidance from the medical literature on the understanding of weight stigma and institutional patient language preferences to inform the development of a future weight management program for patients with overweight and obesity in a single health system context [3, 6, 9, 48]. Our findings highlight the benefits of systematically gathering patient perspectives related to their experiences of stigma and preferences for care within a single healthcare setting. While health system data often focuses on general patient satisfaction, data collection tailored to specific population-relevant experiences or conditions (e.g., obesity, racism, mental health, etc.) can yield important, more detailed information. Similar to other populations, food insecurity increased with each weight class and delaying or canceling care due to health care-associated weight bias also increased with each weight class [13, 49]. Our findings also highlight the benefits of integrating qualitative with quantitative data to provide additional patient insights into weight management experiences and preferences within our health system. This was particularly evident in the qualitative results regarding health system barriers ("Health System Frustration" and "Clinical Environment" themes), the need for additional provider education/training ("Distrust" and "Empathetic Communication" themes), and preferred treatment options ("Intensive Program or Subspecialist Care," "Medication Assisted Treatment," and "Technologic Supports").

## Conclusion

Obesity is a complicated, individualized condition that requires integrated interventions from the community, health system, clinic bedside, and beyond to yield effective, person-centered weight management care [7]. The mixed methods results from this survey reinforce many insights from the literature and add nuance to patient preferences of communication approach by weight class and sex. Our qualitative findings further detailed emotional hinderances, perceptual hinderances, and perceived areas of helpfulness that patients from our health system recommended to improve weight management care in our specific context. Given the continued rise in obesity prevalence, we recommend a health systems approach by bringing interdisciplinary teams together to critically assess their institutional weight management care, informing future context-specific improvement initiatives that emphasize addressing weight stigma and enhancing the equity of their care [7, 43].

## Supporting information

**S1 File. NRC OHSU obesity guidelines survey.**
(DOCX)

## Acknowledgments

The authors of this project would like to thank the survey respondents from OHSU who provided their time and insights into their individual lived experience regarding this topic. We also appreciate NRC Health's collaboration throughout this process, and OHSU's institutional support in recognizing the need to work at a health systems level to improve the weight management-focused care that we provide to our patients. This work was conducted while Dr. Kane was an Internal Medicine resident at OHSU. Dr. Kane is currently a National Clinician

Scholar in the Duke Clinical and Translational Science Institute (CTSI). Through Dr. Kane's affiliation, this project described herein was supported by the Duke CTSI. The content is solely the responsibility of the authors and does not necessarily represent the official views of the Duke CTSI.

## Author Contributions

**Conceptualization:** Ryan M. Kane, Selvi B. Williams, Kimberly Reynolds, Abby Kincanon, Marcy R. Hager, Craig McDougall, Jonathan Q. Purnell, Patricia A. Carney.

**Data curation:** Ryan M. Kane, Selvi B. Williams, Kimberly Reynolds, Abby Kincanon, Marcy R. Hager, Jonathan Q. Purnell, Patricia A. Carney.

**Formal analysis:** Ryan M. Kane, Abby Kincanon, Patricia A. Carney.

**Investigation:** Ryan M. Kane, Selvi B. Williams, Abby Kincanon, Marcy R. Hager, Craig McDougall, Jonathan Q. Purnell, Patricia A. Carney.

**Methodology:** Ryan M. Kane, Abby Kincanon, Marcy R. Hager, Craig McDougall, Jonathan Q. Purnell, Patricia A. Carney.

**Project administration:** Ryan M. Kane, Abby Kincanon, Marcy R. Hager, Craig McDougall.

**Resources:** Ryan M. Kane, Abby Kincanon, Marcy R. Hager, Patricia A. Carney.

**Software:** Ryan M. Kane, Abby Kincanon, Patricia A. Carney.

**Supervision:** Ryan M. Kane, Selvi B. Williams, Kimberly Reynolds, Abby Kincanon, Marcy R. Hager, Craig McDougall, Jonathan Q. Purnell, Patricia A. Carney.

**Validation:** Ryan M. Kane, Abby Kincanon, Patricia A. Carney.

**Visualization:** Ryan M. Kane, Abby Kincanon, Patricia A. Carney.

**Writing – original draft:** Ryan M. Kane, Selvi B. Williams, Kimberly Reynolds, Marcy R. Hager, Craig McDougall, Jonathan Q. Purnell, Patricia A. Carney.

**Writing – review & editing:** Ryan M. Kane, Selvi B. Williams, Kimberly Reynolds, Abby Kincanon, Marcy R. Hager, Craig McDougall, Jonathan Q. Purnell, Patricia A. Carney.

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
