## [Decision Letter · Decision Letter 0]

31 Jul 2024

PONE-D-24-07772Patient perceived weight stigma and patient-centered language use preferences: a cross-sectional mixed methods quality improvement project in a large academic medical centerPLOS ONE

Dear Dr. Kane,

Thank you for submitting your manuscript to PLOS ONE. After careful consideration, we feel that it has merit but does not fully meet PLOS ONE’s publication criteria as it currently stands. Therefore, we invite you to submit a revised version of the manuscript that addresses the points raised during the review process.

We look forward to receiving your revised manuscript.

Kind regards,

Rabie Adel

Academic Editor

PLOS ONE

Additional Editor Comments:

Dear Dr. Kane and Co-Authors,

Thank you for submitting your manuscript titled "Patient perceived weight stigma and patient-centered language use preferences: a cross-sectional mixed methods quality improvement project in a large academic medical center" to PLOS ONE. We have carefully reviewed your submission and, while we recognize the importance of your research, we have identified several areas that require revision before the manuscript can be considered for publication. Please address the following points in your revised manuscript:

General Concerns

Low Response Rate and Representativeness:

The response rate of 19.2% is relatively low and may introduce response bias. Please discuss this limitation in detail and explore methods to increase response rates, such as follow-up reminders or incentives, in future studies.

Consider discussing the potential non-response bias and how it might have affected your findings.

Generalizability:

The study was conducted in a single academic medical center, limiting the generalizability of the findings. Please acknowledge this limitation explicitly and suggest how future research could address this by including multiple institutions and diverse geographic locations.

Sampling and Data Collection:

The use of an opt-in sample may introduce bias. Discuss the potential implications of this and consider suggesting strategies for obtaining a more representative sample in future studies.

As the study is cross-sectional, it only provides a snapshot in time. Highlight this limitation and suggest the need for longitudinal studies to track changes over time and infer causality.

Quantitative Analysis

Statistical Methods:

Ensure that all assumptions for the statistical tests used (Chi-squared and ANOVA) are met and discuss any limitations related to these methods.

Address whether corrections for multiple comparisons were applied to reduce the risk of Type I errors. If not, please include this in your analysis or discuss why it was not necessary.

Handling of Missing Data:

Describe in detail how missing data were handled. If imputation methods were used, please specify and justify them. Discuss the potential impact of missing data on your results.

Variable Categorization:

The categorization of BMI into discrete classes might oversimplify the relationship between BMI and patient experiences. Consider discussing the potential limitations of this approach and suggest alternative methods for future research.

Survey Instrument:

The five-point scales used might not capture the full complexity of patient experiences. Discuss the limitations of this scale and consider suggesting more nuanced scales for future studies.

Consider including a pilot testing phase for your survey instrument in future research to refine questions and scales.

Potential Misinterpretations, Overestimations, and Underestimations:

Misinterpretation: Be cautious about generalizing findings to broader populations beyond the specific academic medical center studied. Acknowledge this limitation and suggest caution in applying these findings universally.

Overestimation: The reported comfort levels in discussing weight and the impact of institutional guidelines might be overestimated due to response bias and the sample's opt-in nature. Address this potential overestimation in your discussion.

Underestimation: Negative experiences and avoidant behaviors due to weight stigma might be underreported. Discuss how the views of non-respondents or those with more negative experiences might not be fully captured.

Qualitative Analysis

Depth and Rigor of Thematic Analysis:

Provide more detail on the thematic analysis process. Include descriptions of how themes were identified, defined, and refined. Consider including measures of inter-coder reliability to ensure the robustness of the thematic analysis.

Sample Representation:

Discuss the representativeness of the qualitative responses and the potential for self-selection bias. Ensure that the views of non-respondents are considered and discuss how this might affect your findings.

Contextual Factors:

Participant Diversity:

Ensure that the analysis captures the diversity of the patient population, including different demographic groups (e.g., age, gender). Discuss any limitations in this regard and suggest improvements for future research.

Ethical and Data Availability Concerns

Longitudinal and Multi-Institutional Studies:

Suggest conducting longitudinal studies and including multiple institutions to improve the generalizability and applicability of the findings.

Table 1: Demographic and Health Care Characteristics of Survey Population

Gender Percentages Not Adding Up:

Ensure that the percentages of male and female participants add up to 100% (excluding missing data)

The totals for each BMI category should be 100%. If there is missing data, it should be noted explicitly

Age Group Percentages Not Summing to 100%:

The age group percentages for each BMI category should sum to 100%.

Inconsistent P-Values:

Ensure p-values are correctly calculated and reported. ( PHYSICAL ACTIVITY)

Table 3: Patients’ Preferred Communication Styles

Incorrect Percentages for Preferences:

Verify the accuracy of percentages for communication preferences. Ensure that the reported preferences and their percentages are correct and consistent with the narrative.

Conclusion

We believe that addressing these concerns will significantly enhance the quality and impact of your manuscript. We appreciate your attention to these matters and look forward to receiving your revised submission. Should you have any questions or require further clarification, please do not hesitate to contact us.

Thank you for your understanding and cooperation.

Best regards,

Reviewers' comments:

Reviewer's Responses to Questions

**Comments to the Author**

1. Is the manuscript technically sound, and do the data support the conclusions?

Reviewer #1: Yes

Reviewer #2: Partly

Reviewer #3: Yes

2. Has the statistical analysis been performed appropriately and rigorously? 

Reviewer #1: Yes

Reviewer #2: I Don't Know

Reviewer #3: Yes

3. Have the authors made all data underlying the findings in their manuscript fully available?

Reviewer #1: No

Reviewer #2: Yes

Reviewer #3: Yes

4. Is the manuscript presented in an intelligible fashion and written in standard English?

Reviewer #1: Yes

Reviewer #2: No

Reviewer #3: Yes

5. Review Comments to the Author

Reviewer #1: Refer to the attached document for specifics regarding the comments outlined below.

1. Abstracts

Background- Clarify vague phrases. Suggested revision: “Due to the prevalence of obesity and its impact on healthcare, patient-specific context is needed to optimize weight management with an emphasis on reducing healthcare-associated weight stigma.”

Objective Statement - Make concise and specific. Suggested revision: “Our study aimed to explore patient attitudes, behaviours, and institutional factors that affect weight management and identify barriers and facilitators for improving care systematically.”

Methods- Clarify phrasing. Suggested revision: “The study utilised a cross-sectional survey among individuals who opted to complete patient experience surveys after receiving care.”

Conclusion- Underscore the scholarly contribution of findings.

2. Introduction

Complex Sentences- Break down long sentences for clarity. Example revision: “Managing overweight and obesity as chronic diseases is challenging due to their multifactorial nature, influenced by individual (e.g., genetic predisposition, co-existing medical conditions), psychological (e.g., weight stigma, behavioral), and societal factors (e.g., social determinants of health).”

Specificity- clarify terms. Include specific examples of “context-specific clinical and community-level interventions.”

Evidence - Provide examples and specific studies to support claims about negative provider attitudes and behaviors, and cite specific patient narratives from qualitative studies.

Rationale- Ground rationale in existing literature, citing recent studies linking weight stigma to adverse health outcomes, and specify gaps in current literature.

Theoretical Frameworks - Reference frameworks supporting patient-centred language in healthcare, such as the Patient-Centred Medical Home (PCMH) model for example.

3. Material and Methods

Translation- Specify qualifications of translators and describe the back-translation process.

Pilot Testing - Details on survey instrument pre-testing procedures and provide preliminary results to demonstrate reliability and validity of the survey items.

4. Results

Percentage Formatting- Ensure consistency by placing all percentages within parentheses.

5. Discussion

Clarification- Clarify vague terms like “indirect approach” and provide specific examples.

Terminology- Replace unclear terms like “improvement project” with more specific descriptions like “research project.”

Transitions- Ensure smooth transitions between statements. Sample revision: “Previous research suggests that patients and providers may not always agree on whether weight management has been discussed during visits. Despite this, it is strongly recommended that providers...”

Citations- Ensure consistent and accurate citation formatting (Vancouver style).

Clarity- Revise sentences for clarity. Sample revisions: “Our finding indicates that respondents preferred direct…” and “Qualitative findings…”.

Reviewer #2: In this studym the authors conducted a cross-sectional survey with an existing institutional group of patients through our organization’s contracted survey vendor (NRC Health). The primary aim of this survey was to understand patient experiences at our institution, including their preferences for communication about obesity care, to inform future health system improvement. In addition, we assessed barriers and facilitators for accessing obesity care

This study is interesting; however, I have some comments:

First, is better to use People-First Language, i.e. "individuals with obesity" rather than "obese ", generally, 'obese' should not be used as an adjective.

Please remove “race” and use only ethnic group or ethnicity; since races do not exist, neither biologically nor scientifically. The human by their common origin, belong to the same genetic repertoire; What does exist is genetic diversity in the human species.

In addition to the points referred to in the methodology related to weight, did the authors take into consideration the comorbidities of the individuals? The question is aimed at whether the authors consider whether the different comorbidities (in addition to weight) could modify the perception of the treatment of health personnel.

Regarding to “The survey population included 16,758 current and former adult patients (≥18 years old) seen at OHSU prior to survey fielding”. Why in Table 1 included individuals < 18 years old?

In Table 1, why were individuals categorized into those age strata? is meas, the authors following any classification (like WHO, etc)?

The authors no considered the menopause as a variable that could modified the answer the women in this status? In addition, the authors believe that gender (like homosexual, lesbian, transexual, etc) could influenced the discomfort that individuals with overweight or obesity felt with the health service?

The authors must to analyze the effect of the low response rate of individuals (19.2%)? Could this represent a bias in your results?

Reviewer #3: This manuscript reports results of a cross-sectional patient experience survey fielded at a single academic medical center. The survey sought to identify individual and system-level factors that facilitate or provide barriers to patient weight management. The survey consisted of both quantitative and qualitative (i.e., open-ended questions). The main findings were that while most patients were happy discussing their weight with their providers, there was variation in the way that patients felt most comfortable with this discussion. Qualitative findings added depth and richness to the qualitative findings, and added the direct patient perspective. Overall, this is a well-written paper on an important topic; statistical and qualitative methods were strong and appropriate. However, I have several suggestions for authors to consider in their revision of the paper.

- While the focus on weight stigma is laudable, the study does not take this concept far enough; it comes from a weight-centric paradigm rather than one of weight-neutrality/health at every size. I.e., it assumes that all persons with obesity require or desire treatment, and in places equates weight and health, which should be avoided. The manuscript could benefit from increased inclusion of a weight-neutral perspective, and critical thinking around the whether or not weight management should always be recommended/is always needed for every patient. The literature on the difficulty of weight loss maintenance and the harms of weight cycling is clear; further, evidence suggests that focus on health, rather than weight, improves cardiometabolic risk factors among those in larger bodies. These concepts are not addressed in the paper sufficiently, and are relevant given some of the patient perspectives given in the qualitative findings.

- Please ensure to use patient-first and non-stigmatizing language throughout. While this is mostly achieved, there is still some language remaining that should be revisited (e.g., use of the term healthy/normal weight, obese patients).

- Response % was low (19.2%), as noted n the limitations; are there EHR data that would allow the authors to compare respondents with non-respondents for socio-demographic characteristics?

- The focus in the first paragraph on other factors that affect health other than fatness is welcomed; however, this paragraph might be strengthened by mentioning additionally the complex causes of obesity beyond “lifestyle factors.” This would strengthen the argument for multi-level interventions, since not all obesity can be (or perhaps should be) cured simply by eating less and moving more.

- The survey-based patient engagement in the development of these guidelines is laudable. Although there is nothing that can be changed about the process at this time, it is unfortunate that there were no patient or fat advocates on the institutional guideline team; this could be a considering moving forward.

- BMI was used to operationalize obesity; given that the flaws of this measure are widely known, and it is not actually recommended by the AMA as a sole measure of obesity, the limitations of this measure should be discussed in the manuscript.

- The results of Table 2 are interesting, and support what we’ve seen in other studies regarding provider weight bias and assumptions about those in larger bodies (e.g., that they don’t engage in physical activity or healthy eating), as well as sub-standard treatment (e.g., delayed or cancelled care). These attitudes are also reflected by patients themselves in the qualitative results. A discussion of provider bias and the importance of addressing this issue should be included in the discussion.

- There are also interesting sex-based differences in findings, especially with regards to patient communication preferences, which may be worth exploring in the discussion with regards to differences in experiences of stigma and internalized bias between males and females.

- Many of the quotations presented from the qualitative work could potentially be interpreted as reflecting internalized bias on the part of the patient. The role of internalized bias on weight management and healthy behavior engagement should be discussed in the discussion section.

6. PLOS authors have the option to publish the peer review history of their article (what does this mean?). If published, this will include your full peer review and any attached files.

Reviewer #1: No

Reviewer #2: **Yes: **Patricia Canto

Reviewer #3: No

---

## [Author Response · Author response to Decision Letter 0]

13 Sep 2024

Dear PLOS ONE Editorial Team and Reviewers, 

Thank you for your thorough review and opportunity to resubmit our manuscript entitled: “Patient perceived weight stigma and patient-centered language use preferences: a cross-sectional mixed methods quality improvement project in a large academic medical center.” We have worked to incorporate your feedback into our manuscript to the best of our ability. 

Please see the below direct responses to all reviewer feedback. All reviewer feedback has been pasted below to facilitate ease of identification of our responses. We look forward to hearing from you soon regarding our revised manuscript.

Kind Regards,

Ryan M. Kane, MD, MPH

Medical Instructor and National Clinician Scholar

Division of General Internal Medicine, Department of Medicine

Duke University School of Medicine 

Duke Clinical and Translational Science Institute

710 W Main St., Durham, NC 27701

ryan.kane@duke.edu

i Manuscript Formatting Guidelines: 

All headings must be written in sentence case (capitalize only the first word of the heading, the first word of the subheading, and any proper nouns). 

>>Response:

Thank you for this formatting insight. We have updated all manuscript headings accordingly. 

1. Abstracts: 

Background 

[Page 2, line 44]: “Given this healthcare system challenge, patient-specific context is needed to optimize weight management with an emphasis on reducing health care-associated weight stigma.” 

The phrase "Given this healthcare system challenge" is somewhat vague and could be clarified. Instead, you could say “Due to the prevalence of obesity and its impact on healthcare……….”. 

>>Response:

Thank you for this suggestion, we have updated this language based on your suggested phrasing to improve specificity: “Due to the rising prevalence of obesity and its impact on healthcare, patient-specific…”

[Page 2, line 46]: “We sought to identify attitudes, behaviors, and institutional systems or structures that hinder and facilitate patients’ weight management to inform future systematic improvements”. 

The sentence is long and could be more concise and specific about the objectives of the study. This could be phrased as “Our study aimed to explore patient attitudes, behaviours, and institutional factors that affect weight management and identify barriers and facilitators for improving care systematically”. 

>>Response:

Thank you for this suggestion, we have updated this language to enhance clarity based on your suggested phrasing: “Our study aimed to explore patient attitudes, behaviors, and institutional factors that affect weight management to identify barriers and facilitators for systematically improving care.”

Material and methods. 

[Page 2, line 48]: “A cross-sectional survey of individuals opted-in to complete patient experience surveys after receiving care”- The phrasing is somewhat unclear. 

Authors may need to clarify this sentence as follows: “The study utilised a cross-sectional survey among individuals who opted to complete patient experience surveys after receiving care……….”. 

>>Response:

Thank you for this suggestion, we have updated this language to enhance clarity based on your suggested phrasing to “This study used a cross-sectional survey among individuals who previously opted-in to complete patient experience surveys after receiving care in a large U.S. academic medical center.”

Conclusion. 

[Page 2, line 61 - 65]: While this study provides practical implications for healthcare systems aiming to improve weight management and reduce stigma, it is essential for the authors to articulate a novel scholarly contribution of their findings in the existing body of knowledge in this field. 

>>Response:

We appreciate the need to clarify our study’s unique contribution to the field whilst staying within the 300-word count maximum of the abstract. We believe our new abstract conclusion addresses this well [line 61-65]:

“Findings from our large single institution cohort expand on the existing weight stigma literature by identifying patient language preferences and healthcare experiences according to patient weight class and sex. Given the potential impact of understanding context-specific patient language use preferences to reduce weight stigma, we recommend other healthcare systems use a similar process to develop institutional plans that address weight stigma as part of a coordinated systems approach for weight management.”

2. Introduction: 

[Page 3, line 70 - 76]: “The management of overweight and obesity as a chronic disease is complicated by the multifactorial impact that individual factors (e.g., genetic and co-occurring medical conditions), psychological factors (e.g., weight stigma and behavioral), and societal factors (e.g., social drivers of health) have upon individuals with overweight or obesity.[2-5] Given this concerning trend and treatment complexity, national and international organizations have recommended a multi-level, multi-modal approach with healthcare institutions integrating context-specific clinical and community-level interventions through a chronic care model.[3, 6, 7]” 

[Page 3, line 70 - 73]: The first sentence is quite long and could be precise by breaking down into more digestible parts. Suggested line should be phrased succinctly as follows: “Managing overweight and obesity as chronic diseases is challenging due to their multifactorial nature, influenced by individual (e.g., genetic predisposition, co-existing medical conditions), psychological (e.g., weight stigma, behavioral), and societal factors (e.g., social determinants of health). [2-5]”. 

>>Response:

Thank you for this suggested modification. We have made this more succinct, as follows: “Managing obesity as chronic disease is challenging due to its multifactorial nature — influenced by individual factors (e.g., genetic and co-occurring medical conditions), psychological factors (e.g., weight stigma and behavioral), and societal factors (e.g., social drivers of health).”

[Page 3, line 73 - 76]: The second sentence should specific which “national and international organizations have made these recommendations” and provide clarification in terms of what is “context-specific clinical and community-level 

interventions through a chronic care model” to clarify for audience (readers) who may not be familiar with these terms/phrases. 

>>Response:

Thank you for offering insights into points to clarify here. I have added additional detail on the institutions and the chronic care model. “Given this concerning trend and treatment complexity, the National Academies of Sciences, Engineering, and Medicine and World Health Organization have recommended multi-level, multi-modal approaches for healthcare institutions to work within their individual clinical and community environments to enhance the prevention of and treatment for obesity, using a chronic care model.[3, 6, 7] The chronic care model highlights critical elements for health care systems to enhance the provision of high-quality chronic disease care across patient, health system, and community spheres of influence.[8]”

[Page 3, line 78 - 85]: “Studies assessing patient and provider communication when discussing weight management have revealed that many patients with obesity experience explicit and implicit stigma when interacting with healthcare providers and systems.[8] Moreover, a recent review modeled how negative provider attitudes and behaviors, such as enacted stigma, threatening environmental cues, and stereotypical behaviors towards patients with obesity can strain the patient-clinician relationship, resulting in adverse patient outcomes (e.g., delayed/avoidant care seeking behaviors, decreased adherence to physician recommendations, and lower likelihood of weight loss).[9, 10]” 

Authors may need to provide examples supporting this assertion “negative provider attitudes and behaviors,” to specify instances of these behaviors (e.g., the use of stigmatizing language, focusing solely on weight without considering broader health issues). 

Furthermore, consider citing specific studies or reports that illustrate patient experiences of stigma rather than general broad statements (e.g., provide details from qualitative studies with the data that describe patient narratives). 

>>Response:

Due to the amount of information in this sentence, I broke it up for the sake of clarity and added additional references from qualitative studies to support these statements:

“Moreover, a recent review modeled how negative provider attitudes and behaviors can strain the patient-clinician relationship, resulting in adverse patient outcomes (e.g. delayed/avoidant care seeking behaviors, decreased adherence to physician recommendations, and lower likelihood of weight loss).[10, 11] These negative provider attitudes and behaviors include (1) enacted stigma (behaviors resulting from negative attitudes like decreased patient-centered communication due to providers’ belief that patients with obesity are less likely to adhere to treatments), (2) threatening environmental cues (clinical spaces and equipment that are not accommodating to patients of all sizes), and stereotypical behaviors (providers demonstrating less respect towards patients with obesity due to perceptions of laziness).[10] Review of qualitative analyses identified related emergent themes surrounding (1) verbal/non-verbal communication of stigma, (2) the impact of weight stigma upon care provision, and (3) weight stigma’s impact upon systemic barriers to care.[12]”

[Page 4, line 102 - 106]: “To assess this, we conducted a cross-sectional survey with an existing institutional group of patients through our organization’s contracted survey vendor (NRC Health). The primary aim of this survey was to understand patient experiences at our institution, including their preferences for communication about obesity care, to inform future health system improvement.” 

Authors should provide a clear rationale grounded in existing literature or studies on why understanding patient perceptions of weight stigma and preferences for language use is crucial for healthcare outcomes. For instance, cite most recent studies rather than official websites that link weight stigma to adverse health behaviors or outcomes (e.g., delayed care, non-adherence to treatment). 

Likewise, authors ought to specify the gaps in current literature or evidence that the study aims to address. This could include gaps in understanding how different communication approaches affect patient engagement in weight management. 

>>Response:

Thank you for this recommendation. We have bolstered the literature base in this paragraph and provided additional context regarding health system gaps. “Due to the literature on weight stigma’s deleterious effects for patient engagement in preventive care, healthcare avoidance, biased physician decision making, poor patient-provider communication, and increased primary care switching; we sought to understand how patients with overweight or obesity in our health system perceived their prior care experiences.[10, 15-18] Though prior literature demonstrated weight stigma-focused analyses of specific cohorts, we were unaware of prior health systems who had published a similar institutional approach to better understand the direct language use preferences and care experiences of their own patient population.”

[Page 4, line 96 - 102]: “The institutional guideline development process aimed to improve the management of people living with overweight and obesity through developing guidelines, standards, and policies that promote optimal patient health. A multi-disciplinary team of experts from across the institution convened for 3 years to review the relevant evidence base and develop guidelines appropriate for this institutional context. During this process, we identified the importance of including how patients with overweight or obesity in our health system perceived their prior encounters for weight management care, particularly when discussing weight management with their primary care providers.” 

Provide empirical evidence or theoretical frameworks that support the significance of patient-centred language in healthcare interactions, particularly in the context of obesity management. This could include references to primary studies. The study could draw upon frameworks like the Patient-Centered Medical Home (PCMH) model which focuses comprehensive and coordinated care that is centred on patient preferences, needs, and values. Read: https://doi.org/10.1186/s13643-022-02132-x and consider utilizing patient-centred approaches in obesity management, which take into account patient perceptions and preferences. Read: https://doi.org/10.1016/j.jand.2022.01.004 and https://doi.org/10.1159/000496183

>>Response:

Thank you for these helpful insights. I have reviewed your citations and included highlighted the importance of patient-centered communication, motivational interviewing, and evidence-based obesity treatment grounded in the literature (including the NAM framework by Dietz, et al.).

“The goal of this is to ensure evidence-based, patient-centered obesity care is available to all patients in our health system to reduce the experience of weight stigma, improve health outcomes, reduce costs, and limit patient harm. Much of this starts with provider communication and the use of patient-centered language or motivational interviewing to compassionately engage patients in a conversation about their weight.[15-17]”

3. Material and Methods: 

[Page 5, line 135 - 136]: “The survey was also translated into Spanish for primary care clinics that primarily serve Spanish-speaking patients and distributed online through specific clinic partnerships.” 

Authors must specify who performed the initial translation into Spanish and the qualifications of the translators (e.g., bilingual experts in healthcare communication) 

and describe the back-translation process, where the translated survey was translated back into English to verify accuracy and consistency. 

Pilot testing is crucial to identify and resolve any issues with survey comprehension, clarity, and appropriateness before full-scale implementation. Provide details if applicable on how the pilot testing was conducted. Furthermore, authors should provide preliminary or baseline results from the validation process to demonstrate the reliability and validity of survey items. 

>>Response:

Thank you for these comments. I have adjusted the text accordingly to respond to these comments:

“After the survey was developed by national survey experts from NRC Health, it was iteratively reviewed by the institutional weight management guideline subcommittee for survey development as well as the multi-stakeholder guideline development team to enhance face and content validity prior to its administration via email between September 29, 2021 and October 5, 2021.[23] The survey was also translated into Spanish by certified bilingual healthcare interpreters for primary care clinics”

Regarding requested information from “pilot testing” and “preliminary or baseline results from the validation process,” we have revised the limitations section to indicate that we did not conduct rigorous test-retest analyses to provide validity data, but we are confident that our iterative expert and stakeholder review process resulted in a survey that produced accurate results. Given the initial focus if this project as a quality improvement project and our use of a national survey vendor (NRC Health) contracted with our institution, pilot testing beyond face and content validity was not planned as part of NRC Health’s standard process when working with health systems on projects such as ours. We did previously acknowledge this limitation in the limitations section of our Discussion though have bolstered this section, as follows: “The overall survey had very few previously validated questions to assess patient preferences and was not pre-tested among a pilot population prior to full dissemination; though it was developed in partnership with a national third-party vendor and informed by iterative multi-disciplinary clinician expert meetings to establish face and content validity. Future partnershi

---

## [Decision Letter · Decision Letter 1]

1 Oct 2024

PONE-D-24-07772R1Patient perceived weight stigma and patient-centered language use preferences: a cross-sectional mixed methods quality improvement project in a large academic medical center

PLOS ONE

Dear Dr. Kane,

Thank you for submitting your manuscript to PLOS ONE. After careful consideration, we feel that it has merit but does not fully meet PLOS ONE’s publication criteria as it currently stands. Therefore, we invite you to submit a revised version of the manuscript that addresses the points raised during the review process.

Many thanks to the authors for this interesting study. In addition to the comments raised already by the reviewers that have been responded to, I have a number of suggested comments/edits for your consideration:

-
The title is a bit misleading—seeing quality improvement immediately suggests a specific improvement methodology was used, but this paper is not about the QI, but rather a study led by a subset of the QI participants. I’d recommend revising the title to something like, “Patient perceived weight stigma and patient-centered language use preferences: a cross-sectional mixed methods study in a large academic medical center”-
Should line 71 be genetics and should line 72 be behaviours, not behavioural?-
Related to my comment about the title, it very likely makes sense to describe this as its own study, briefly acknowledging that this was developed by a subgroup participating within a wider QI intervention. Please reduce the overall description of the QI (it’s also not clear whether it was about developing guidelines, implementing guidelines, or both) and the text from 93 to 106 should be revised and put into the methods. Then please add a paragraph to the conclusion of the introduction to highlight why it’s important to get a good understanding of perceived stigma/experiences receiving care and the importance of preferred patient-centred communication—basically, please make it crystal clear for the reader why this is a useful study to do. -
The details about OHSU being an MD-granting med school, etc. on lines 113 and 114 can be removed. -
Can you please provide some detail on sample size calculations to ensure you had sufficient numbers to carry out the analyses. -
Please define PCP (line 174)-
How are under 18s included in Table 1 when the survey was only sent to adults?-
Please remove the two longer quotations from your discussion—there shouldn’t be results here.-
A significant limitation of the qual aspect is that the open-ended questions seem a bit leading. What if an overweight/obese patient hadn’t felt uncomfortable?  Please also don’t refer to the limitations as being of your “improvement project”, as you don't actually speak to the QI intervention in this paper (e.g. none of your results are about this, nor should they be, as per my previous comment). **Please note that reviewer 3's comments have not been included in this email, please review the previous correspondence with these included and ensure these are also responded to.**

We look forward to receiving your revised manuscript.

Kind regards,

Tara Tancred, PhD

Academic Editor

PLOS ONE

Journal Requirements:

Reviewers' comments:

Reviewer's Responses to Questions

**Comments to the Author**

1. If the authors have adequately addressed your comments raised in a previous round of review and you feel that this manuscript is now acceptable for publication, you may indicate that here to bypass the “Comments to the Author” section, enter your conflict of interest statement in the “Confidential to Editor” section, and submit your "Accept" recommendation.

Reviewer #1: All comments have been addressed

2. Is the manuscript technically sound, and do the data support the conclusions?

Reviewer #1: Yes

3. Has the statistical analysis been performed appropriately and rigorously? 

Reviewer #1: Yes

4. Have the authors made all data underlying the findings in their manuscript fully available?

Reviewer #1: No

5. Is the manuscript presented in an intelligible fashion and written in standard English?

Reviewer #1: Yes

6. Review Comments to the Author

Reviewer #1: Dear authors, I would to commended you for addressing the revisions thoroughly in enhancing the overall clarity of the manuscript for it reach scientific rigor.

Abstract:

[line 48]: Use only “Methods” and remove material. In the same line, Authors must rephrase the following sentence for clarity and indicate the type of mixed methods design adopted: “A cross-sectional survey of individuals opted-in to complete patient experience surveys after receiving care in a large academic medical center in the United States of America (U.S.).”

Suggested revision if the study was either explanatory or exploratory sequential or concurrent or embedded in nature.: “This study adopted an explanatory sequential mixed methods design, using cross-sectional survey, with a sample of individuals who opted in to complete patient experience surveys after receiving care at a large academic medical center in the United States (U.S.).”

Introduction:

[Page 4, line 104 – 106]: Dear Authors, please rephrase the following statement to better resonate with the scope and specific focus of the study. “The primary aim of this survey was to understand patient experiences at our institution, including their preferences for communication about obesity care, to inform future health system improvement. In addition, we assessed barriers and facilitators for accessing obesity care”

Suggested sentence that retains focus of the study: “The primary aim of this survey was to understand patient experiences at our institution, with a specific focus on patient weight stigma and their preferences for patient-centered language use in communication about obesity care, to inform future health system improvement”

Results:

[Page 7, line 169]: Dear Authors, I suggest that you remove parentheses in the following phrase for consistency and please note that it is only percentages ought to be within the parentheses: “Nearly 9,000 patients (8,714 of 16,758) …..”

[Page 7, Line 170] “……completed the survey (response rate=19.2%)” have to be rephrased as follows: ……completed the survey accounting for/or representing the response rate of (19.2%) and …..

7. PLOS authors have the option to publish the peer review history of their article (what does this mean?). If published, this will include your full peer review and any attached files.

Reviewer #1: No

---

## [Author Response · Author response to Decision Letter 1]

22 Oct 2024

Dear Dr. Tancred, PLOS ONE Editorial Team, and Reviewers, 

Thank you for calling out the comments we missed during our prior revision, as well as the opportunity to submit a new modified version of our paper, now entitled: “Patient perceived weight stigma and patient-centered language use preferences: a cross-sectional mixed methods analysis conducted in a large academic medical center.” 

We have now incorporated all reviewer feedback into the revised manuscript. We again apologize for any confusion we caused. We have now clarified all reviewer responses below, including Reviewer #1’s additional (second) feedback, the initial email body Reviewer feedback, and the attached Reviewer feedback.

All reviewers’ feedback has been pasted below to facilitate ease of identification of our responses. Given the length of these reviews, we have placed page-breaks to denote the sections between reviewer feedback blocks. We look forward to hearing from you soon regarding our revised manuscript and appreciate your continued consideration of this important work.

Kind Regards,

Ryan M. Kane, MD, MPH

Medical Instructor and National Clinician Scholar

Division of General Internal Medicine, Department of Medicine

Duke University School of Medicine 

Duke Clinical and Translational Science Institute

710 W Main St., Durham, NC 27701

ryan.kane@duke.edu

>> START OF DR. TANCRED RESPONSES <<

- The title is a bit misleading—seeing quality improvement immediately suggests a specific improvement methodology was used, but this paper is not about the QI, but rather a study led by a subset of the QI participants. I’d recommend revising the title to something like, “Patient perceived weight stigma and patient-centered language use preferences: a cross-sectional mixed methods study in a large academic medical center”

>>Response: Thank you for this feedback. Actually, this project is part of a larger effort designed to improve the quality of physician patient communication, for which we are using PDSA as a QI Framework because of the value of iterative assessments and modifications. Because of this, our IRB approved this project as QI; thus, we are unable to use the word “study” in the title. We have revised the project description to clarify this point and retitled the manuscript to focus on the “survey” rather than the larger QI project as such: “Patient perceived weight stigma and patient-centered language use preferences: a cross-sectional mixed methods analysis conducted in a large academic medical center.” Please see the ‘Methods’ section for additional details regarding the overall initiative.

- Should line 71 be genetics and should line 72 be behaviours, not behavioural?

>>Response: Great catch!!! We have made these changes accordingly: “genetics” and “behaviors.”

- Related to my comment about the title, it very likely makes sense to describe this as its own study, briefly acknowledging that this was developed by a subgroup participating within a wider QI intervention. Please reduce the overall description of the QI (it’s also not clear whether it was about developing guidelines, implementing guidelines, or both) and the text from 93 to 106 should be revised and put into the methods. Then please add a paragraph to the conclusion of the introduction to highlight why it’s important to get a good understanding of perceived stigma/experiences receiving care and the importance of preferred patient-centered communication—basically, please make it crystal clear for the reader why this is a useful study to do. 

>>Response: Thank you for this comment. As mentioned above, we cannot characterize this effort as a study, due to our IRB determination. We have, however, edited the final 2 paragraphs of the ‘Introduction’ to better highlight the known impacts of weight stigma: “…deleterious effects for patient engagement in preventive care, healthcare avoidance, biased physician decision making, poor patient-provider communication, and increased primary care switching.” (Additional information included prior to and after this quoted text). 

We have also moved information regarding the larger guideline development process to the second paragraph of the ‘Methods’ and clarified both the importance and the focus of this paper on the survey in the last paragraph of the introduction, as suggested. 

- The details about OHSU being an MD-granting med school, etc. on lines 113 and 114 can be removed. 

>>Response: We have removed this text, as suggested. 

- Can you please provide some detail on sample size calculations to ensure you had sufficient numbers to carry out the analyses. 

>>Response: Again, this is not a research study. Thus, we did not work with the survey vendor to calculate sample size related to an anticipated effect size and power. Rather these analyses were exploratory. We have worked to clarify the exploratory nature of this survey in ‘Methods’ (See page 7).

- Please define PCP (line 174)

>>Response: Apologies for missing this. We have defined PCP accordingly as “primary care provider (PCP)”

- How are under 18s included in Table 1 when the survey was only sent to adults?

>>Response: Thank you for this question. We have clarified this in the last two sentences of the ‘Methods’ section under ‘Study population & survey administration,’ as “Some parents of patients under 18 were included in the Community Insights Program and were kept in the analyses given the exploratory nature of this survey and the overall guideline development focus upon both pediatric and adult obesity” (See page 7). 

- Please remove the two longer quotations from your discussion—there shouldn’t be results here.

>>Response: Thank you for this feedback. We have revised the text accordingly to remove the in-text quotations.

- A significant limitation of the qual aspect is that the open-ended questions seem a bit leading. What if an overweight/obese patient hadn’t felt uncomfortable? Please also don’t refer to the limitations as being of your “improvement project”, as you don't actually speak to the QI intervention in this paper (e.g. none of your results are about this, nor should they be, as per my previous comment). 

>>Response: We respectfully disagree that the open-ended questions were leading - Please see the S1 File supplemental information to review the entire survey and its logic. We already indicated that only respondents that answered the immediately preceding, related multiple-choice questions as ‘somewhat uncomfortable’ or ‘very uncomfortable’ were able to respond to these two of the three open-ended questions. We have slightly revised the final paragraph of the ‘Methods’ ‘Study setting and survey development’ section to highlight this survey logic.

As mentioned, this is not a research study. That said, given the focus of this paper is on the survey data and not the larger improvement project, We have adjusted the language to say “limitations of this analysis” instead of “improvement project” in the ‘Discussion.’

>> END OF DR. TANCRED RESPONSES<< 

>>START OF REVIEWER #1’S SECOND EMAIL BODY REVIEW RESPONSES<<

Reviewer #1: 

Dear authors, I would like to commend you for addressing the revisions thoroughly in enhancing the overall clarity of the manuscript for it reach scientific rigor.

Abstract:

[line 48]: Use only “Methods” and remove material. In the same line, Authors must rephrase the following sentence for clarity and indicate the type of mixed methods design adopted: “A cross-sectional survey of individuals opted-in to complete patient experience surveys after receiving care in a large academic medical center in the United States of America (U.S.).”

Suggested revision if the study was either explanatory or exploratory sequential or concurrent or embedded in nature.: “This study adopted an explanatory sequential mixed methods design, using cross-sectional survey, with a sample of individuals who opted in to complete patient experience surveys after receiving care at a large academic medical center in the United States (U.S.).”

>>Response: Thank you for your subsequent review of this manuscript. We have made your suggested changes altered slightly to adhere to the abstract word count restrictions: “This cross-sectional survey adopted a concurrent mixed methods design…”

Introduction:

[Page 4, line 104 – 106]: Dear Authors, please rephrase the following statement to better resonate with the scope and specific focus of the study. “The primary aim of this survey was to understand patient experiences at our institution, including their preferences for communication about obesity care, to inform future health system improvement. In addition, we assessed barriers and facilitators for accessing obesity care”

Suggested sentence that retains focus of the study: “The primary aim of this survey was to understand patient experiences at our institution, with a specific focus on patient weight stigma and their preferences for patient-centered language use in communication about obesity care, to inform future health system improvement”

>>Response: Thank you for this suggested revision. We changed it accordingly, as we believe you have stated this more concisely. 

Results:

[Page 7, line 169]: Dear Authors, I suggest that you remove parentheses in the following phrase for consistency and please note that it is only percentages ought to be within the parentheses: “Nearly 9,000 patients (8,714 of 16,758) …..”

>>Response: We have adjusted this accordingly to remove the parenthetical: “Of the 16,758 patients in the total sample, 8,714 opened…”

[Page 7, Line 170] “……completed the survey (response rate=19.2%)” have to be rephrased as follows: ……completed the survey accounting for/or representing the response rate of (19.2%) and …..

>>Response: We have adjusted this accordingly to restructure the parenthetical: “…survey, yielding a response rate of (19.2%)…”

>>END OF REVIEWER #1’S SECOND EMAIL BODY REVIEW RESPONSES<< 

>> START OF INITIAL EMAIL BODY REVEIWER RESPONSES <<

Additional Editor Comments:

General Concerns

Low Response Rate and Representativeness:

The response rate of 19.2% is relatively low and may introduce response bias. Please discuss this limitation in detail and explore methods to increase response rates, such as follow-up reminders or incentives, in future studies.

Consider discussing the potential non-response bias and how it might have affected your findings.

>>Response: Thank you for this suggestion. We have added additional information regarding this in the ‘Discussion’ that expands on our prior statement: “…there was a moderate response rate at 19.2% with a possibility of response bias towards those with personal experience of weight stigma, or in contrast, those who experienced less stigma and were more comfortable responding…” with several strategies to enhance the response rate to reduce non-response bias: “seeking information via phone call, focus group, in person or paper survey, providing multi-modal reminder messages, and including study staff with representative backgrounds for direct contact may be more likely to elicit responses from some patients than an online survey. Additionally, including representative patients in project method and survey development is an increasing and important practice for improving access to representative data. All of these methods to enhance the diversity of the sample, in addition to the provision of compensation for survey completion, would likely have the added benefit of enhancing survey response rates, mitigating the potential impact of non-response bias.”

Generalizability:

The study was conducted in a single academic medical center, limiting the generalizability of the findings. Please acknowledge this limitation explicitly and suggest how future research could address this by including multiple institutions and diverse geographic locations.

>>Response: Thank you for this comment, we have expanded on what was previously written as follows: “Although the single institution focus was the intent, generalizations should be made with caution, and geographic and cultural differences should be accounted for within any individual healthcare system. Future multi-institution surveys across a range of geographic locations would enhance the generalizability of this information, though would be less specific to the individual health system context.”

Sampling and Data Collection:

The use of an opt-in sample may introduce bias. Discuss the potential implications of this and consider suggesting strategies for obtaining a more representative sample in future studies.

As the study is cross-sectional, it only provides a snapshot in time. Highlight this limitation and suggest the need for longitudinal studies to track changes over time and infer causality.

>>Response: Thank you for this suggestion, we have added to the limitations section as such: “The opt-in nature of this online survey may similarly impact non-response bias, as compared to direct surveying of patients as they engaged with care. Relatedly, the cross-sectional nature of this survey provides a single snapshot in time for these patients and relies on recall of their past experiences. Longitudinally surveying patients, particularly as health system improvements are implemented, would provide more meaningful insights into the direct impact of weight stigma and language use preferences with enhanced confidence in these associations.”

Quantitative Analysis

Statistical Methods:

Ensure that all assumptions for the statistical tests used (Chi-squared and ANOVA) are met and discuss any limitations related to these methods.

>>Response: We have added basic information on this to the ‘Methods’ ‘Data analyses’ section building on the exiting first sentence: “For quantitative analyses, descriptive statistics, including frequencies, means, medians, standard deviations, ranges, and percentiles were calculated to assess the shape of the data…All assumptions for Chi-squared and ANOVA tests were assessed and met prior to performing any analyses. These statistical tests were chosen based on the presence of categorical and continuous data with the goal to assess overall effects rather than individual interactions across the independent participant survey responses.”

Address whether corrections for multiple comparisons were applied to reduce the risk of Type I errors. If not, please include this in your analysis or discuss why it was not necessary.

>>Response: Apologies for not including this in the first version, but we did include a Bonferroni correction to account for multiple comparisons, specifically to reduce the risk of Type I errors in analyses. 

Handling of Missing Data:

Describe in detail how missing data were handled. If imputation methods were used, please specify and justify them. Discuss the potential impact of missing data on your results.

>>Response: Thank you for this comment. We have added the following to the ‘Methods:’ “We report the sample sizes for each individual question’s answer choices by category to display the missingness from each item and denote overall missingness in Table 1. No additional methods were employed to statistically address missingness.” & the ‘Discussion:’ “Individual question item non-response may accentuate the potential impact upon the generalizability of these findings, though overall missingness was low and transparently displayed for each item through the displayed sample sizes in each table.”

Variable Categorization:

The categorization of BMI into discrete classes might oversimplify the relationship between BMI and patient experiences. Consider discussing the potential limitations of this approach and suggest alternative methods for future research.

>>Response: Thank you, we have added this to the ‘Discussion’ accordingly: “Our focus was to identify variability in patient experiences and preferences by widely recognized weight classifications, as clinicians discuss weight more often among patients with higher BMIs...Future studies may explore patient experiences and language use preferences keeping BMI as a continuous outcome in a regression model to enhance the statistical detail of using BMI over weight class categories.”

Survey Instrument:

The five-point scales u

---

## [Decision Letter · Decision Letter 2]

8 Nov 2024

Patient perceived weight stigma and patient-centered language use preferences: a cross-sectional mixed methods analysis conducted in a large academic medical center

PONE-D-24-07772R2

Dear Dr. Kane,

We’re pleased to inform you that your manuscript has been judged scientifically suitable for publication and will be formally accepted for publication once it meets all outstanding technical requirements.

Kind regards,

Tara Tancred, PhD

Academic Editor

PLOS ONE

Additional Editor Comments (optional):

Many thanks to the authors for this extensively revised manuscript, which is reading very clearly now. On the reviewer's point on the use of race rather than ethnic group--I do fundamentally believe that ethnic group or ethnicity is more appropriate, but it seems clear from your survey instrument that it was in fact "race" that was used in the instrument, and therefore that is what you should report on. However, as this moves through to publication, you may consider acknowledging this as a limitation, as rethinking/revising such language is something we, as a research health community, need to do better at. 

Reviewers' comments:

Reviewer's Responses to Questions

**Comments to the Author**

1. If the authors have adequately addressed your comments raised in a previous round of review and you feel that this manuscript is now acceptable for publication, you may indicate that here to bypass the “Comments to the Author” section, enter your conflict of interest statement in the “Confidential to Editor” section, and submit your "Accept" recommendation.

Reviewer #2: All comments have been addressed

Reviewer #3: All comments have been addressed

2. Is the manuscript technically sound, and do the data support the conclusions?

Reviewer #2: Yes

Reviewer #3: Yes

3. Has the statistical analysis been performed appropriately and rigorously? 

Reviewer #2: Yes

Reviewer #3: Yes

4. Have the authors made all data underlying the findings in their manuscript fully available?

Reviewer #2: Yes

Reviewer #3: Yes

5. Is the manuscript presented in an intelligible fashion and written in standard English?

Reviewer #2: Yes

Reviewer #3: Yes

6. Review Comments to the Author

Reviewer #2: The authors responded satisfactorily to most of my comments.

However, the fact that people of Caucasian origin use the term "race" consistently does not mean that it is right (regardless of the fact that you point out many references that use it that way), so you should change race to ethnic group.

Reviewer #3: (No Response)

7. PLOS authors have the option to publish the peer review history of their article (what does this mean?). If published, this will include your full peer review and any attached files.

Reviewer #2: No

Reviewer #3: No

---

## [Editor Report · Acceptance letter]

18 Nov 2024

PONE-D-24-07772R2 

PLOS ONE

Dear Dr. Kane, 

I'm pleased to inform you that your manuscript has been deemed suitable for publication in PLOS ONE. Congratulations! Your manuscript is now being handed over to our production team.

Kind regards, 

on behalf of

Dr. Tara Tancred 

Academic Editor

PLOS ONE